



# Validation of aerosol backscatter profiles from Raman lidar and ceilometer using balloon-borne measurements

Simone Brunamonti[1], Giovanni Martucci[1], Gonzague Romanens[1], Yann Poltera[2], Frank G. Wienhold[2] Alexander Haefele[1] and Francisco Navas-Guzmán[1]

[1]Federal Office of Meteorology and Climatology (MeteoSwiss), Payerne, Switzerland
[2]Swiss Federal Institute of Technology (ETH), Zürich, Switzerland

*Correspondence to:* Francisco Navas-Guzmán (francisco.navasguzman@meteoswiss.ch)

**Abstract.** Remote sensing measurements by light detection and ranging (lidar) instruments are fundamental for the monitoring of altitude-resolved aerosol optical properties. Here, we validate vertical profiles of aerosol backscatter coefficient ($\beta_{aer}$) measured by two independent lidar systems using co-located balloon-borne measurements performed by Compact Optical Backscatter Aerosol Detector (COBALD) sondes. COBALD provides high-precision in-situ measurements of $\beta_{aer}$ at two wavelengths (455 and 940 nm). The two analyzed lidar systems are the research Raman Lidar for Meteorological Observations (RALMO) and the commercial CHM15K ceilometer (Lufft, Germany). We consider in total 17 RALMO and 31 CHM15K profiles, co-located with simultaneous COBALD soundings performed throughout the years 2014-2019 at the MeteoSwiss observatory of Payerne (Switzerland). The RALMO (355 nm) and CHM15K (1064 nm) measurements are converted to respectively 455 nm and 940 nm using the Angstrom exponent profiles retrieved from COBALD data. To account for the different receiver field of view (FOV) angles between the two lidars (0.01-0.02°) and COBALD (6°), we derive a custom-made correction using Mie-theory scattering simulations. Our analysis shows that both RALMO and CHM15K achieve a good agreement with COBALD measurements in the boundary layer and free troposphere, up to 6 km altitude, and including fine structures in the aerosol's vertical distribution. For altitudes below 2 km, the mean ± standard deviation difference in $\beta_{aer}$ is + 6 % ± 40 % (+ 0.005 ± 0.319 Mm$^{-1}$ sr$^{-1}$) for RALMO – COBALD at 455 nm, and + 13 % ± 51 % (+ 0.038 ± 0.207 Mm$^{-1}$ sr$^{-1}$) for CHM15K – COBALD at 940 nm. The large standard deviations can be at least partly attributed to atmospheric variability effects, associated with the balloon's horizontal drift with altitude (away from the lidar beam) and the different integration times of the two techniques. Combined with the high spatial and temporal variability of atmospheric aerosols, these effects often lead to a slight altitude displacement between aerosol backscatter features that are seen by both techniques. For altitudes between 2-6 km, the absolute standard deviations of both RALMO and CHM15K decrease (below 0.13 and 0.16 Mm$^{-1}$ sr$^{-1}$, respectively), while their corresponding relative deviations increase (often exceeding 100% COBALD of the signal). This is due to the low aerosol content (i.e. low absolute backscattered signal) in the free troposphere, and the vertically decreasing signal-to-noise ratio of the lidar measurements (especially CHM15K). Overall, we conclude that the $\beta_{aer}$ profiles measured by the RALMO and CHM15K lidar systems are in good agreement with in-situ measurements by COBALD sondes up to 6 km altitude.



## 1. Introduction

Aerosol particles are ubiquitous in the atmosphere, and play a key role in multiple processes that affect weather and climate. They absorb and scatter the incoming and outgoing radiation, which affects the Earth's radiative budget (direct effect), and interact with cloud formation processes, influencing their microphysical properties and lifetime (indirect effect) (e.g., Haywood
and Boucher, 2000). Atmospheric aerosols are one of the largest sources of uncertainty in current estimates of anthropogenic radiative forcing (Bindoff et al., 2013).

Among the most significant causes of this uncertainty is the high variability, in space and time, of the aerosol's concentration, composition and optical properties. Remote sensing instruments, such as light detection and ranging (lidar) systems, represent an optimal tool for the monitoring of altitude-resolved aerosol optical coefficients (backscatter and extinction), especially in
the planetary boundary layer (PBL) (e.g., Amiridis et al., 2005; Navas-Guzmán et al., 2013). Lidar networks like EARLINET (www.earlinet.org) and E-PROFILE (www.eumetnet.eu/e-profile), comprising several hundreds of single-wavelength (including ceilometers) and multi-wavelength (Raman) lidars, provide a comprehensive database of the horizontal, vertical and temporal distribution of aerosols over Europe (e.g., Bösenberg et al., 2003; Pappalardo et al., 2014; Sicard et al., 2015).

Lidar instruments offer the advantages of vertically-resolved measurements and continuous operation in time, but are subject
to a number of intrinsic uncertainties of this technique. Single-wavelength elastic backscatter lidars are limited by the fact that only one signal is measured, while the returning intensity is determined by two parameters (backscatter and extinction). Hence, an a-priori assumption on the aerosol extinction-to-backscatter ratio (the so-called 'lidar ratio') is necessary for the calculation of the aerosol backscatter profiles (e.g., Collis and Russel, 1976). Additionally, the retrieval at low altitudes is particularly challenging because of the incomplete geometric overlap between the incoming beam and the receiver's field of view (e.g.,
Wandinger and Ansmann, 2002; Weitkamp, 2005; Navas-Guzmán et al., 2011). The comparison of aerosol backscatter profiles from 19 elastic backscatter lidars in EARLINET found deviations within 10% in the PBL compared to more advanced Raman lidar measurements (Matthais et al., 2004).

Multi-wavelength Raman lidars allow the independent measurement of aerosol backscatter and extinction as functions of altitude, by the detection of a pure molecular backscatter signal in addition to the elastic backscatter (Ansmann et al., 1990; 1992).
However, the retrieval procedure is complex and prone to uncertainties, in particular for extinction. It involves the calculation of the derivative of the logarithm of the ratio between the atmospheric number density and the lidar received power, which generally requires complex data handling techniques to isolate the signal from statistical fluctuations (Pappalardo et al., 2004). The comparison of different aerosol backscatter retrieval algorithms between 11 Raman lidar systems in EARLINET, using synthetic input data, showed deviations between them up to 20% for altitudes below 2 km (Pappalardo et al., 2004). This calls
for careful validation studies against independent in-situ measurements, as we perform in this work.

In-situ instruments are characterized by higher precision and signal-to-noise ratio compared to remote sensing measurements, but are typically limited by low spatial and temporal coverage. Altitude-resolved in-situ measurements of aerosol optical properties can be achieved by various platforms including aircrafts, unmanned aerial vehicles (UAVs), and meteorological balloons.





Specifically, balloon-borne measurements of aerosol backscatter are typically used to investigate high-altitude cirrus clouds (e.g., Khaykin et al., 2009; Cirisan et al., 2014) and aerosol layers in the upper troposphere and stratosphere (e.g., Rosen and Kjome, 1991; Vernier et al., 2015; Brunamonti et al., 2018), which are not accessible by aircrafts and UAVs. The aim of this paper is to use balloon-borne measurements of aerosol backscatter in the lower troposphere to validate the retrievals of aerosol

backscatter coefficient by one co-located Raman lidar and one co-located ceilometer.

The instruments and data used for the comparison are introduced in detail in Section 2. The method of comparison, including the derivation of a field of view (FOV) correction from idealized Mie-theory scattering simulations, is described in Section 3. The results of the comparison are discussed in Section 4, and the conclusions summarized in Section 5.

## 2. Observations

We analyze vertical profiles of aerosol backscatter coefficient ($\beta_{aer}$) measured by two remote sensing instruments, namely one research Raman lidar system and one commercial ceilometer, and in-situ (balloon-borne) measurements performed by aerosol backscatter sondes. The three instruments and measuring techniques are introduced in Sections 2.1-2.2, and their main characteristics are summarized in Table 1. All data were collected at the MeteoSwiss Aerological Observatory of Payerne, Switzerland (46.82° N, 6.95° E), located at an elevation of 491 m above sea level (asl), between January 2014 and October 2019.

The selection of the dataset considered for the statistical comparison is discussed in Section 2.3.

### 2.1. Remote sensing measurements

RALMO (Raman Lidar for Meteorological Observations) is a research Raman lidar system developed by EPFL Lausanne in collaboration with MeteoSwiss (Dinoev et al., 2013), operational in Payerne since 2008 and part of the EARLINET network. It uses a Nd:YAG laser source, which emits pulses of 8 ns duration at wavelength 355 nm and frequency of 30 Hz. The laser

beam divergence is 120 µrad and the mean energy per pulse 400 mJ. The receiving system consists of four telescopes with 30 cm parabolic mirrors, with equivalent total aperture of 60 cm and field of view (FOV) angle of 200 µrad. Optical fibers connect the telescope mirrors with two polychromators, which allow to isolate the rotational-vibrational Raman signals of nitrogen and water vapor (wavelengths 386.7 nm and 407.5 nm, respectively) and the pure rotational Raman lidar signals (around 355 nm). The rotational-vibrational signals are used to derive water vapor profiles (Brocard et al., 2013; Hicks-Jalali et al., 2019; 2020),

while the pure rotational signals are used for temperature, aerosol backscatter and aerosol extinction coefficients (e.g., Dinoev et al., 2010; Martucci et al., 2018). The optical signals are detected by photomultipliers and acquired by a transient recorder system (Brocard et al., 2013). Aerosol backscatter measurements from RALMO were recently used to characterize hygroscopic growth during mineral dust and smoke events (Navas-Guzmán et al., 2019). Here we derive the RALMO $\beta_{aer}$ at 355 nm from the ratio between the elastic and inelastic signal, as described in Navas-Guzmán et al. (2019).





The CHM15K-Nimbus (hereafter CHM15K) ceilometer is a single-wavelength elastic backscatter lidar manufactured by Lufft, Germany (Lufft, 2019), installed in Payerne since 2012, and member of E-PROFILE. It uses a Nd:YAG narrow-beam micro-chip laser emitting 1 ns pulses at wavelength 1064 nm and repetition rate between 5-7 Hz, and a receiver FOV of 450 μrad. It supports a range up to 15 km with first overlap point at 80 m and full overlap reached at 800 m above the station (Hervo et al.,

2016). CHM15K is employed as a cloud height sensor and for the automatic detection of boundary layer height (Poltera et al., 2017) and it was used for the characterization of aerosol hygroscopic properties (Navas-Guzmán et al., 2019). Here we derive $\beta_{aer}$ at 1064 nm from the CHM15K elastic signal using the Klett inversion technique (Klett, 1981). For consistency within the statistical comparison, we assume a constant lidar ratio equal to 50 sr for all profiles.

## 2.2. In-situ measurements

COBALD (Compact Optical Backscatter Aerosol Detector) is a light-weight (500 g) aerosol backscatter detector for balloon-borne measurements developed at ETH Zürich, based on the original prototype by Rosen and Kjome (1991). Using two light emitting diodes (LEDs) as light sources and a photodiode detector with FOV of 6°, COBALD provides high-precision in-situ measurements of aerosol backscatter at wavelengths of 455 nm (blue visible) and 940 nm (infrared). COBALD was originally developed for the observation of high-altitude clouds, such as cirrus (e.g., Brabec et al., 2012; Cirisan et al., 2014) and polar

stratospheric clouds (Engel et al., 2014), while recently it was proven able to detect and characterize aerosol layers in the upper troposphere and lower stratosphere (e.g., Vernier et al., 2015; 2018; Brunamonti et al., 2018). Here, for the first time we use COBALD measurements for the analysis of lower tropospheric aerosols.

For each balloon sounding, the COBALD sonde is connected to a host radiosonde via their XDATA interface (e.g., Wendell and Jordan, 2016) to transmit the data to the ground station. The average ascent rate of the balloon is set to around 5 m/s, which

combined with a measurement frequency of 1 Hz, provides a vertical resolution of approximately 5 m. Typical balloon burst altitude is about 35 km. Due to the high sensitivity of its photodiode detector, COBALD sondes can be only deployed during night-time. Hence, all soundings analyzed here were started at approximately 23:00 UTC. More than 100 COBALD soundings were performed in Payerne since 2009, supported by SRS-C34 radiosondes by MeteoLabor, Switzerland (MeteoLabor, 2010) until December 2017, and RS41-SGP radiosondes by Vaisala, Finland (Vaisala, 2017) since January 2018.

The COBALD measurements are typically expressed as backscatter ratio (BSR) at 455 and 940 nm, obtained by dividing the total measured signal by its molecular contribution, which is computed from the atmospheric extinction according to Bucholtz (1995) and using air density derived from the radiosonde measurements of temperature and pressure (e.g., Cirisan et al., 2014). Here we further derive $\beta_{aer}$ from the COBALD BSR assuming a molecular extinction-to-backscatter ratio of $8\pi/3$ sr.



## 2.3. Dataset

Over their operational periods, the RALMO and COBALD systems were subject to various technical and design modifications, which affected their characteristics and performances. In particular, the currently used COBALD 940 nm LED was introduced in January 2014, replacing the older 870 nm LED (e.g., Brabec et al., 2012), while the pure rotational Raman acquisition board

of RALMO was replaced, from a Licel system to the faster FAST ComTec P7888 (FastCom, Germany), in August 2015 (see Martucci et al., 2018). For consistency, we consider in this work only the time periods following these changes, i.e. the current versions of RALMO and COBALD up-to-date. Therefore, we analyze the years 2014-2019 for the CHM15K validation (58 total COBALD soundings), and the years 2016-2019 for the RALMO validation (34 total soundings: note that no simultaneous RALMO-COBALD soundings are available between August and December 2015).

Out of all the available COBALD soundings, we exclude those with simultaneously missing or incomplete (up to at least 6 km altitude) lidar profiles. This can be due to instrumental failures, maintenance interventions, or forbidding weather conditions (e.g. thick low clouds, fog or precipitation) at the time of the COBALD sounding. In particular, we reject from the comparison all profiles for which a precise calibration of the lidar signal cannot be achieved. The calibration of lidar (as well as COBALD) measurements involves the normalization of the signal to a reference value in a 'clean region' (i.e. the lowest aerosol concen-

tration along the profile), usually found in the upper troposphere. If no lidar signal is measured in this region of altitudes, which is typically the case in the presence of thick low clouds, or if the signal-to-noise ratio above the cloud is so low that the signal cannot be properly calibrated, then the profile is excluded from the comparison. After a careful selection, we obtain 17 simultaneous calibrated profiles of RALMO and COBALD and 31 of CHM15K and COBALD, which are used for the statistical comparison. The list of corresponding dates is given by Table S1 in Supplementary material.

## 3. Method of comparison

For a proper comparison of the remote sensing and balloon-borne measurements, a number of methodological and technical differences between the lidar and COBALD instruments need to be taken into account. Here we discuss our approach towards spatial and temporal variability issues (Section 3.1), wavelength homogenization (Section 3.2), and correction of effects related to the different receiver FOVs (Section 3.3).

## 3.1. Spatial and temporal variability

For each COBALD sounding, we retrieve simultaneous RALMO and CHM15K profiles with vertical resolution of 30 m and integration time of 30 min (roughly corresponding to 10 km of balloon ascent time). Since all COBALD sondes were launched at 23:00 UTC, the integration time window chosen for all profiles and both lidars is 23:00-23:30 UTC. To obtain a dataset with consistent vertical levels, the COBALD measurements (with vertical resolution ≈ 5 m) are averaged in altitude bins of 30





m, matching the vertical grid of the lidars. For the statistical comparison we consider in total 174 vertical levels, covering the altitude interval from 800 m asl to 6 km asl. We only select measurements from $\approx$ 300 m above the ground station in order to minimize the effect a possible incomplete overlap of the lidar systems in the lower part of the profiles. Note that all altitude levels given in the following are meant as altitude asl, unless differently specified.

Along with the COBALD backscatter data, the temperature, pressure and relative humidity (RH) measurements from the host radiosonde are averaged to the same altitude levels. The temperature and pressure profiles are used for the computation of the atmospheric molecular extinction, as described in Section 2.2. The RH measurements are used to reject in-cloud data points. In-cloud aerosol backscatter measurements are typically much larger (up to three orders of magnitude) compared to clear-sky (i.e. aerosol-only) conditions, and characterized by high spatial and temporal variability. Therefore, we exclude from the com-

parison all data points with RH > 90%. Such a highly conservative criterion is chosen in order to avoid as well cloud edge regions, which can lead to large biases in the statistical comparison.

   A fundamental difference between the remote sensing and balloon sounding techniques is that lidars measure at every altitude the vertical air column directly above their laser beam, while the balloon sondes are subject to a horizontal drift with altitude, dictated by the atmospheric wind field. Therefore, in presence of wind shear, the two instruments may not measure the same

air mass at every altitude. The distance between the balloon sonde and the lidar beam generally increases with altitude, and is strongly dependent on the atmospheric wind profile at the time of measurement. Figure 1 shows the trajectories of all balloon soundings analyzed in our comparison for the period 2016-2019, as function of altitude (0.8-6 km). The distance between the lidar and the sondes ranges between roughly 0-5 km up to 2 km altitude, and may exceed 10 km at 4 km altitude.

   In addition, the two techniques differ in terms of measurement times. Namely, while the lidar profiles are integrated 30 min in

time, COBALD provides instantaneous measurements at 1 s resolution (reduced to 6 s after averaging to 30 m intervals). The combination of balloon drift with altitude and different integration times, coupled with the high spatial and temporal variability of aerosol optical properties, can lead to discrepancies between the remote sensing and in-situ measurement which are not due to instrumental issues, but rather to atmospheric variability effects. In particular, this may result in the smoothing or slight displacement in altitude between aerosol backscatter features (especially thin layers) which are seen by both techniques. Such

effects are often observed in our dataset (see Section 4.1) and are not corrected in the statistical comparison, hence they contribute to increasing the standard deviation of the results.

## 3.2. Wavelength conversion

To compare $\beta_{aer}$ at different wavelengths ($\lambda$) measured by the different instruments, it is necessary to account for the spectral dependency of aerosol backscatter. This can be done using the Angstrom law (Equation 1), which describes the spectral de-

pendency of $\beta_{aer}$ between two wavelengths ($\lambda_0$ and $\lambda$) as function of the Angstrom exponent (AE) at every altitude ($z$). The AE is an intensive property of the aerosol that, under certain assumptions on the particle's size distribution, can be used as a semi-





quantitative indicator of particle size (e.g., Njeki et al., 2012; Navas-Guzmán et al., 2019). Through Equation 1 we convert the lidar profiles into the COBALD wavelengths, so they can be quantitatively compared.

$$\beta_{aer}(\lambda, z) = \beta_{aer}(\lambda_0, z) \cdot \left(\frac{\lambda}{\lambda_0}\right)^{-AE(z)} \qquad \text{(Equation 1)}$$

Thanks to its high signal-to-noise ratio and two operating wavelengths, COBALD allows to characterize the backscatter spectral ratio (between 455 nm and 940 nm) at every altitude, including regions of low aerosol load (e.g., Brunamonti et al., 2018). Conversely, the signal-to-noise ratio of remote sensing instruments (in our case especially CHM15K) decreases with altitude, and the AE derived from lidar measurements is typically characterized by large statistical fluctuations in the free troposphere. Therefore, here we choose to retrieve the $AE(z)$ profiles from COBALD data. To minimize the uncertainty associated with the conversion, we couple each lidar with the closest COBALD channel in terms of wavelength. Hence, the RALMO profiles at 355 nm (ultraviolet) are converted to 455 nm and compared to the COBALD blue visible channel, and the CHM15K profiles at 1064 nm (infrared) are converted to 940 nm and compared to the COBALD infrared channel.

Using the AE from COBALD is equivalent to assuming that the spectral behavior of the aerosols between 455-940 nm can be extrapolated to the slightly broader interval of 355-1064 nm, which is justified by the small difference between the wavelengths that are compared. A number of sensitivity tests using different assumptions have been conducted, revealing that small changes in AE have a small effect on the results (e.g., less than 2% change in $\beta_{aer}$ for a 10% change in AE, for 455 nm and AE $\approx$ 1). In particular, the entire statistical comparison discussed in Section 4.2.1, using AE profiles from COBALD, was repeated using AE profiles derived from RALMO and CHM15K measurements, and assuming a constant AE = 1 for all altitudes. The results are displayed in Figure S1 in Supplementary material and show no relevant variations.

### 3.3. Field of View (FOV) correction

Besides their wavelengths, the COBALD and lidar systems differ in terms of field of view (FOV) of their respective receivers. RALMO and CHM15K use highly focused laser beams, and consequently have narrow FOVs (200 μrad and 450 μrad, respectively, corresponding to 0.01-0.02°), while COBALD's photodiode detector has a macroscopic FOV of 6° (see Table 1). Considering that the Mie-scattering phase function, i.e. the distribution of scattered light with angle by a spherical particle, has a local maximum in the backward direction (180°), it follows from its wider FOV that COBALD will measure less backscattered radiation (namely, the average intensity between 174°-180°) compared to the lidars ($\approx$ 180°).

To quantify this effect, we performed idealized Mie-theory scattering simulations using the optical model by Luo et al. (2003). We assume a single lognormal size distribution of aerosol particles characterized by mode radius $R_m$, number concentration $N$, fixed width ($\sigma = 1.4$), and refractive index 1.4. Then, the BSR of this population is computed both assuming the phase function





value at 180°, corresponding to the lidar observations (FOV ≈ 0°), and taking the average of the phase function between angles 174°-180°, corresponding to the COBALD measurements (FOV = 6°). The results are presented in Figure 2.

Figure 2a shows the simulated aerosol backscatter ratio (i.e. BSR − 1) at 455 nm (blue) and 940 nm (red) calculated assuming FOV ≈ 0° (solid) and FOV = 6° (dashed), as function of $R_m$ for the interval 10 nm - 4 μm, and $N = 10^3$ cm$^{-3}$. As expected, the
simulations show that for all mode radii the COBALD BSR is lower than the BSR measured by the lidar instruments (Figure 2a). Figure 2b shows the lidar-to-COBALD ratio of BSR − 1 (ratio of solid-to-dashed curves in Figure 2a), i.e. the correction factor required to compensate for this effect, for 455 and 940 nm as function of $R_m$ (note that $\beta_{aer}$ is proportional to BSR − 1, so that this ratio corresponds to the correction factor for $\beta_{aer}$). For the considered interval of mode radii, the correction factors vary between approximately 1-1.5 and show a non-linear dependency on $R_m$, with a local maximum near $R_m \approx 800$ nm ($\lambda =$
455 nm) and $R_m \approx 1.6$ μm ($\lambda = 940$ nm). This complex optical behavior needs to be corrected. Note that the correction factors in Figure 2b are independent of $N$, unlike the BSR in Figure 2a.

To account for the size-dependency in Figure 2b, we use the AE as an indicator of particle size, and develop a parametrization of the correction factors based on the AE measured from COBALD. Figure 2c shows AE between 455-950 nm calculated from the Mie simulations, as function of $R_m$. The AE decreases non-monotonically with mode radius and exhibits the characteristic
Mie oscillations in the range of approximately 20 nm - 1 μm (Figure 2c). More in detail, we observe that AE > 1.5 corresponds to small particles ($R_m < 75$ nm), AE < 0.8 to large particles ($R_m > 1.16$ μm), while 0.8 < AE < 1.5 corresponds to 75 nm < $R_m$ < 1.16 μm, but in this intermediate range the change of AE with $R_m$ is not monotonic (Figure 2c), hence a one-to-one correspondence cannot be established. To simplify this behavior, we choose to parametrize the correction factors within the three fixed intervals of AE just introduced, and for each interval of AE we take the average correction factor in the corresponding
interval of $R_m$. Hence, for all measurements with 0.8 < AE < 1.5 we apply the average correction factors between 75 nm - 1.16 μm (namely, 1.23 at 455 nm, 1.10 at 940 nm), for AE < 0.8 the average correction factors between 1.16 - 4 μm (1.29 at 455 nm, 1.28 at 940 nm), and for AE > 1.5 we do not apply any correction (both correction factors ≈ 1 for $R_m < 75$ nm). The resulting FOV correction as function of AE is shown in Figure 2d.

The correction shown in Figure 2d is applied to all COBALD measurements in the statistical comparison. Since, for every AE,
the correction factors are larger for 455 nm than for 940 nm (Figure 2d), the FOV correction will affect the RALMO comparison more than for CHM15K. We note that, due to the variability of AE observed in our dataset (see Figure S2 in Supplementary material), the middle interval of the correction (0.8 < AE < 1.5) accounts for the large majority of data points in the PBL, AE > 1.5 typically corresponds to free tropospheric background measurements, which are unaffected by the correction, while values of AE < 0.8, corresponding to very large particles, are rarely encountered in our dataset. The effect of the FOV correc-
tion on individual profiles and the statistical comparison is discussed further in the next section.



## 4. Results

In this section we present the results of our analysis. Before the statistical comparison (Section 4.2), we discuss the comparison of two selected individual profiles (Section 4.1), highlighting the effect of the FOV correction. Note that two additional examples of individual profiles can be found in Supplementary material (Figures S3-S4).

### 4.1. Comparison of individual profiles

To illustrate the main characteristics of the observed $\beta_{aer}$ profiles and the effect of the FOV correction, we select two individual cases corresponding to the COBALD soundings made on 12 July 2018 and 4 September 2018. Figure 3 shows an overview of these measurements, including vertical profiles of $\beta_{aer}$ (at different $\lambda$) by RALMO, COBALD and CHM15K (panels a, d), AE derived from COBALD measurements (panels b, e), plus the temperature and RH profiles measured by the radiosonde (panels

c, f), for the altitude interval of 0.8-6 km.

The case of 12 July 2018 (Figure 3, top row) shows a typical profile with top of PBL at about 2.2 km altitude (see temperature inversion, Panel c), characterized by a sharp decrease with altitude in $\beta_{aer}$ and RH, plus a thin ($\approx$ 400 m) isolated aerosol layer around 3 km altitude (note the higher AE compared to the PBL, suggesting finer particles: Panel b). Inside the PBL, the vertical structure of $\beta_{aer}$ observed by COBALD is qualitatively well reproduced by both RALMO and CHM15K, despite an evident

altitude displacement (of about 60 m) of the top-of-PBL decrease in $\beta_{aer}$ between the COBALD and lidar profiles (Figure 3a). This is most likely an effect of the atmospheric variability issues discussed in Section 3.1. Indeed, considering that COBALD crosses the PBL around the beginning of the lidar integration time window, a downward displacement in top of PBL altitude (as inferred from the $\beta_{aer}$ profiles) in the remote sensing data is consistent with the lowering of PBL altitude during nighttime reported by Poltera et al. (2017). A similar feature can be seen in Figure S3d in Supplementary material.

On 4 September 2018 (Figure 3, bottom row) a more complex aerosol vertical distribution is observed, with decreasing $\beta_{aer}$ with altitude until 2 km, and a thick aerosol layer between 2.5-3.5 km altitude. Again, the vertical structure of $\beta_{aer}$ observed by COBALD is qualitatively well reproduced by both remote sensing products throughout the entire analyzed altitude range, including both aerosol layers inside and above the PBL. In this case, no significant altitude displacement is observed between the $\beta_{aer}$ features of the COBALD and remote sensing profiles (Figure 3d).

Figure 4 shows the results of the quantitative comparison for the two cases just discussed, meaning the $\beta_{aer}$ profiles obtained after converting the lidar wavelengths (355 to 455 nm and 1064 to 940 nm) and applying the FOV correction to the COBALD measurements. In particular, Figure 4 shows vertical profiles of $\beta_{aer}$ at 455 nm from RALMO and COBALD (Panels a, e), $\beta_{aer}$ at 940 nm from CHM15K and COBALD (Panels c, g), and their respective differences ($\Delta\beta_{aer}$) at 455 nm (Panels b, f) and 940 nm (Panels d, h), for 12 July 2018 (Panels a-d) and 4 September 2018 (Panels e-h). The COBALD $\beta_{aer}$ and $\Delta\beta_{aer}$ profiles are

shown both before (dashed lines) and after (solid lines) the FOV correction.



The FOV correction significantly improves the agreement between RALMO and COBALD measurements. Before the FOV correction (dashed lines), the RALMO profiles are characterized by a systematic high bias with respect to COBALD, of about 0.2 Mm$^{-1}$ sr$^{-1}$ for $z < 2$ km (Figures 4a-b, 4e-f). After the FOV correction (solid lines), which increases the COBALD $\beta_{aer}$ by a factor of 1.23 in this region of altitudes (see Figure 2d and AE profiles in Figure 3b), the discrepancy with RALMO is drasti-

cally reduced, and the profiles are in good agreement within ± 0.1 Mm$^{-1}$ sr$^{-1}$ (Figures 4b, 4f). Note that the $\Delta\beta_{aer}$ discrepancy associated with the altitude displacement on 12 July 2018 increases after the FOV correction (Figure 4b).

As already noted in Section 3.3 (Figure 2d), the effect of the FOV correction on the CHM15K comparison is smaller. Particularly, we observe that for 4 September 2018 (Figure 4g-h) the FOV correction leads to a slight improvement in agreement with COBALD ($\approx 0.05$ Mm$^{-1}$ sr$^{-1}$), whereas on 12 July 2018 (Figure 4c-d) it slightly increases the discrepancy with COBALD.

Due to the empirical implementation of the FOV correction, with many assumptions and simplifications involved (e.g. single-mode size distribution, coarse parameterization in AE-space, etc.), it is to be expected that for individual sounding the magnitude of our correction might be underestimating or overestimating the true effect of the different FOVs. Nevertheless, the FOV correction systematically improves the statistical comparison between COBALD and both RALMO and CHM15K, as will be discussed in the next section (see Tables 2-3).

**4.2. Statistical comparison**

Here we discuss the results of the statistical comparison for the dataset introduced in Section 2.3, consisting of 17 simultaneous RALMO vs. COBALD profiles (Section 4.2.1) and 31 CHM15K vs. COBALD profiles (Section 4.2.2).

**4.2.1. RALMO vs. COBALD**

Figure 5 shows all data points of the RALMO – COBALD difference ($\Delta\beta_{aer}$ at 455 nm) as function of altitude, both expressed
in absolute backscatter coefficient units (Panel a) and in percent units relative to the COBALD signal (Panel b), after the FOV correction was applied to all COBALD measurements. The mean and mean ± standard deviation profiles of $\Delta\beta_{aer}$ are shown by thick solid and thin dashed black lines, respectively. As discussed in Section 3.1, to avoid in-cloud measurements, we only consider data points with RH < 90% (according to the radiosonde measurements).

RALMO and COBALD measurements are on average in good agreement over the entire altitude range (0.8-6 km), yet signif-
icant discrepancies can occur in single profiles, and the standard deviation is not constant with altitude. For $z > 2.5$ km, typically corresponding to the free troposphere (i.e. above the PBL), the absolute differences between RALMO and COBALD are small (Figure 5a), while their relative differences are large (Figure 5b). This is mostly due to the low aerosol content, hence the low absolute backscattered signal, in 'clean' free-tropospheric air masses. The absolute $\Delta\beta_{aer}$ differences for $z > 2.5$ km are smaller





than 0.1 Mm$^{-1}$ sr$^{-1}$ for the majority of data points (Figure 5a), yet their relative discrepancies often exceed ± 100% of the signal, and the mean $\Delta\beta_{aer}$ profile varies between 0-30 % (Figure 5b).

For $z < 2.5$ km, which approximately corresponds to the average top of PBL altitude in our dataset, the discrepancies between RALMO and COBALD are larger in absolute terms (Figure 5a), but smaller in relative terms (Figure 5b) compared to the free troposphere. The mean $\Delta\beta_{aer}$ profile for $z < 2.5$ km stays within ± 0.1 Mm$^{-1}$ sr$^{-1}$ with standard deviation ≈ 0.25 Mm$^{-1}$ sr$^{-1}$, while individual data points rarely exceed ± 0.5 Mm$^{-1}$ sr$^{-1}$ (Figure 5a). In relative terms, this corresponds to an average slight over-estimation of 5-10 % and a standard deviation of ≈ 40 % (Figure 5b). A large fraction of this high variability can be attributed to the atmospheric variability effects discussed in Section 3.1, namely, the smearing or displacement in altitude of $\beta_{aer}$ features that are seen by both techniques, due to the balloon's drift with altitude away from the lidar beam, and the different measuring times of the lidar and COBALD techniques (see Figures 3-4, S3-S4).

To quantify the spread of $\Delta\beta_{aer}$, Figure 6 shows frequency of occurrence distribution of the RALMO – COBALD difference, calculated for the altitude intervals of 0-2 km (Panels e-f), 2-4 km (Panels c-d), and 4-6 km (Panels a-b) (note that the first interval only contains data from above 800 m asl, as discussed in Section 3.1). The distributions are calculated both in absolute units, within 40 intervals of 0.1 Mm$^{-1}$ sr$^{-1}$ width between ±2 Mm$^{-1}$ sr$^{-1}$ (left column plots), and in percent units relative to the COBALD signal, within 40 intervals of 10 % width between ±200 % (right column). The mean values and standard deviations of each distribution are noted in every panel.

The frequency of occurrence distributions for $z > 2$ km highlight the small variability in absolute units (Figure 6a, 6c) and large variability in relative units (Figure 6b, 6d) already observed in Figure 5. The absolute standard deviation for 2-6 km altitude is around 0.12 Mm$^{-1}$ sr$^{-1}$ (Figure 6a, 6c), corresponding to up to 95 % of the signal, at 4-6 km (Figure 6b, 6d). For z < 2 km, the average RALMO – COBALD difference is +0.005 Mm$^{-1}$ sr$^{-1}$ (standard deviation 0.319 Mm$^{-1}$ sr$^{-1}$) in absolute units (Figure 6e), and +6 % (standard deviation 40 %) in relative units (Figure 6f). The skewedness of the distributions towards large values is mainly due to a single outlying profile, showing discrepancies larger than +100 % at $z < 2$ km (see Figure 5b).

Finally, Figure 7 shows a correlation (scatter) plot of all COBALD vs. RALMO measurements of $\beta_{aer}$ at 455 nm (Panel a), and two additional frequency of occurrence distributions of $\Delta\beta_{aer}$ (Panels b-c). The scatter plot includes all data points between altitudes 0.8-6 km of the 17 profiles considered for the comparison (blue circles) plus their mean correlation line (thick black line), calculated as the average $\beta_{aer}$ from RALMO for each 0.2 Mm$^{-1}$ sr$^{-1}$ interval of $\beta_{aer}$ from COBALD (Figure 7a). Isolines of 1:1 agreement, ±25 % and ±50 % difference are also indicated by thin black lines. The frequency of occurrence distributions, expressed both in absolute (Panel b) and relative units (Panel c), are calculated from all data points with COBALD $\beta_{aer} > 0.2$ Mm$^{-1}$ sr$^{-1}$. This threshold is set to exclude all free tropospheric samples with negligible aerosol content, which may introduce large relative discrepancies despite of very small absolute differences (see bottom-left corner of Figure 7a).

Figure 7a reveals that the majority of RALMO measurements with $\beta_{aer} > 0.2$ Mm$^{-1}$ sr$^{-1}$ (at all altitudes) lie between a ± 25 % difference with respect to COBALD, whereas for $\beta_{aer} < 0.2$ Mm$^{-1}$ sr$^{-1}$ deviations exceeding ± 100 % are commonly observed. The average correlation line stays within ± 25 % for nearly the entire considered signal range, and approaches a 1:1 agreement between 0.6-1.4 Mm$^{-1}$ sr$^{-1}$ (Figure 7a), which is typically the relevant range of $\beta_{aer}$ at 455 nm in the PBL (e.g., Figures 4, S4).





Considering all data points with $\beta_{aer} > 0.2$ Mm$^{-1}$ sr$^{-1}$, the average RALMO – COBALD difference is – 0.011 Mm$^{-1}$ sr$^{-1}$ (+ 1.7 %) with standard deviation 0.329 Mm$^{-1}$ sr$^{-1}$ (56 %) in absolute (relative) units, respectively (Figure 7b-c).

To summarize the statistical comparison, Table 2 reports mean values and standard deviations of RALMO – COBALD $\Delta\beta_{aer}$ for the three altitude intervals defined in Figure 6 (i.e., 2 km intervals between 0-6 km), and for the ensemble of all measure-

ments with COBALD $\beta_{aer} > 0.2$ Mm$^{-1}$ sr$^{-1}$ as defined in Figure 7b-c. In addition to Figures 6-7, Table 2 shows the same results calculated both before and after the application of the FOV correction, which enables to evaluate its effect on the statistical comparison. The mean $\Delta\beta_{aer}$ decreases from 29 % to 6 % after the FOV correction for $z < 2$ km, and from 26 % to 2 % when considering all measurements with $\beta_{aer} > 0.2$ Mm$^{-1}$ sr$^{-1}$. Above 2 km, the effect of the FOV correction is smaller (e.g., 33 % to 29 % at 4-6 km) due to the prevalence of free tropospheric air masses (with high AE) that are unaffected by the correction (see

Figure 2d). The standard deviations are weakly affected by the FOV correction (e.g., 45 % to 40 % for $z < 2$ km).

### 4.2.2. CHM15K vs. COBALD

Following the same structure of the previous subsection, here we analyze the CHM15K vs. COBALD statistical comparison first in terms of vertical profiles (Figure 8), then frequency of occurrence distributions (Figure 9), and finally scatter plot of all CHM15K vs. COBALD measurements (Figure 10).

Figure 8 shows all data points of $\Delta\beta_{aer}$ at 940 nm for CHM15K – COBALD as function of altitude, both in absolute backscatter coefficient units (Panel a) and percent units relative to the COBALD signal (Panel b). Like in Figure 5, the FOV correction is applied to all COBALD measurements, and only data points with RH < 90 % are considered. Note that the higher density of data points in Figure 8 compared to Figure 5 is due to the larger number of profiles considered for the CHM15K vs. COBALD comparison (31) relative to the RALMO vs. COBALD comparison (17) (see Section 2.3).

In absolute units, the CHM15K measurements are on average in good agreement with COBALD over the entire altitude range (Figure 8a), although their relative differences are characterized by strong statistical fluctuations at high altitudes (Figure 8b). For $z > 2.5$ km, the absolute differences between CHM15K and COBALD are smaller than ± 0.2 Mm$^{-1}$ sr$^{-1}$ for the majority of data points (Figure 8a), while the corresponding relative discrepancies often exceed ±100 % (Figure 8b). This is due to the low absolute backscattered signal and the low signal-to-noise ratio of CHM15K in the free troposphere. We observe that the abso-

lute differences between CHM15K and COBALD (Figure 8a) are typically larger than for RALMO (Figure 5a) at all altitudes, despite $\beta_{aer}$ is smaller at 940 nm than at 455 nm due to its spectral dependency. This highlights the lower signal-to-noise ratio of the CHM15K ceilometer compared to a high-power Raman lidar such as RALMO, and causes the large relative fluctuations of $\Delta\beta_{aer}$ in the free troposphere observed in Figure 8b.

For $z < 2.5$ km, CHM15K shows on average a slight overestimation with respect to COBALD measurements. The mean $\Delta\beta_{aer}$

in the PBL is about +0.04 Mm$^{-1}$ sr$^{-1}$ with standard deviation 0.2 Mm$^{-1}$ sr$^{-1}$ (Figure 8a), corresponding to about +15 % (standard deviation ≈ 50 %) of the COBALD signal (Figure 8b). The large standard deviation can be again partly attributed to atmospheric variability effects, in addition to the lower signal-to-noise ratio of CHM15K discussed above. The slight positive bias





of CHM15K with respect to COBALD for $z < 2.5$ km could be due to minor unsolved geometric overlap issues in the ceilometer's retrieval algorithm, or (more likely) related to the assumption of a constant lidar ratio (50 sr) for all profiles made in the Klett inversion scheme, used for the retrieval of $\beta_{aer}$ from CHM15K (see Section 2.1).

Figure 9 shows the frequency of occurrence distribution of CHM15K – COBALD $\Delta\beta_{aer}$ at 940 nm for the altitude intervals of 0-2 km (Panels e-f), 2-4 km (Panels c-d), and 4-6 km (Panels a-b), expressed both in absolute units (left column) and in percent units relative to the COBALD signal (right column), and calculated as in Figure 6. The frequency of occurrence distributions for $z > 2$ km again highlight the small variability of $\Delta\beta_{aer}$ in absolute units (Figure 9a, 9c) and large variability in relative units (Figure 9b, 9d) in the free troposphere. We observe that the spread of the $\Delta\beta_{aer}$ distributions, and particularly the associated standard deviations, are larger for CHM15K (e.g., 320 % at 2-4 km, Figure 9d; 640% at 4-6 km, Figure 9b) than for RALMO (see Figure 6). For $z < 2$ km (Figure 9e-f), the mean CHM15K – COBALD difference is $+ 0.038$ Mm$^{-1}$ sr$^{-1}$ (+13 %) with standard deviation 0.207 Mm$^{-1}$ sr$^{-1}$ (51 %) in absolute (relative) units, respectively.

Finally, analogously to Figure 7, Figure 10 shows a scatter plot of all COBALD vs. CHM15K measurements of $\beta_{aer}$ at 940 nm (Panel a), plus two additional frequency of occurrence distributions of $\Delta\beta_{aer}$ (Panels b-c). The scatter plot includes all data points between altitudes 0.8-6 km of the 31 profiles considered for the comparison (orange circles), plus the mean COBALD vs. CHM15K correlation profile (thick black line), calculated as in Figure 7. The frequency of occurrence distributions, both in absolute (Panel b) and relative units (Panel c), are calculated from all data points with $\beta_{aer} > 0.1$ Mm$^{-1}$ sr$^{-1}$ (lowered threshold in order to account for the smaller values of $\beta_{aer}$ at 940 nm compared to 455 nm).

Figure 10a shows a general overestimation of CHM15K compared to COBALD measurements in the range of approximately 0.4-1.2 Mm$^{-1}$ sr$^{-1}$, which is consistent with the vertical profiles shown in Figure 8. Most of the data points in this interval are found in the top-left quadrant of the scatter plot, with discrepancies often exceeding +50 % of the COBALD signal, while their mean correlation line stays between 0-25 % (Figure 10a). When all measurements with $\beta_{aer} > 0.1$ Mm$^{-1}$ sr$^{-1}$ are considered, the average CHM15K – COBALD difference is $– 0.001$ Mm$^{-1}$ sr$^{-1}$ (+1.9 %) with standard deviation of 0.281 Mm$^{-1}$ sr$^{-1}$ (43 %) in absolute (relative) units, respectively (Figure 10b-c).

In summary, Table 3 shows mean values and standard deviations of CHM15K – COBALD $\Delta\beta_{aer}$ for the three altitude intervals defined in Figure 9, and for the ensemble of all measurements with COBALD $\beta_{aer} > 0.1$ Mm$^{-1}$ sr$^{-1}$ defined in Figure 10b-c. As in Table 2, here we show the same results both as calculated before and after the application of the FOV correction. The mean CHM15K – COBALD $\Delta\beta_{aer}$ decreases due to the FOV correction from 25% to 13% for $z < 2$ km, and from 16 % to 2% when considering all measurements with $\beta_{aer} > 0.1$ Mm$^{-1}$ sr$^{-1}$. For $z > 2$ km, the effect of the correction is small.

## 5. Conclusions

We have presented the first comparison of lower tropospheric aerosol backscatter coefficient ($\beta_{aer}$) profiles retrieved by remote sensing instruments against independent in-situ measurements. The two analyzed lidar systems, one research Raman lidar (RALMO) and one commercial ceilometer (CHM15K), were validated using simultaneous and co-located balloon soundings



carrying a Compact Backscatter Aerosol Detector (COBALD), performed during the years 2014-2019 at the MeteoSwiss observatory of Payerne, Switzerland. COBALD provides high-precision in-situ measurements of $\beta_{aer}$ at two wavelengths (455 and 940 nm) and is used as the reference instrument. The $\beta_{aer}$ profiles retrieved from RALMO (355 nm) and CHM15K (1064 nm) are converted to respectively 455 nm and 940 nm using the altitude-dependent Angstrom exponent (AE) profiles retrieved

from COBALD data. To account for the different receiver field of view (FOV) angles between the remote sensing instruments (0.01-0.02°) and COBALD (6°), we derived a FOV correction using Mie-theory scattering simulations. The correction factors are parametrized as functions of AE to account for the size-dependency of the solutions (see Figure 2).

The comparison of individual profiles shows that both RALMO and CHM15K achieve a good agreement with COBALD $\beta_{aer}$ measurements in the boundary layer and free troposphere, up to 6 km altitude, including fine structures in the aerosol's vertical

distribution. For altitudes below 2 km, the mean ± standard deviation difference in $\beta_{aer}$ obtained from the statistical comparison of all available profiles is +6 ± 40 % (+0.005 ± 0.319 Mm$^{-1}$ sr$^{-1}$) for RALMO – COBALD at 455 nm, and +13 ± 51 % (+0.038 ± 0.207 Mm$^{-1}$ sr$^{-1}$) for CHM15K – COBALD at 940 nm. The high standard deviations can be largely attributed to atmospheric variability effects related to fundamentally different characteristics of the remote sensing and balloon sounding techniques, as the balloon's horizontal drift with altitude (away from the lidar beam) and the different integration times. Combined with the

high spatial and temporal variability of atmospheric aerosols, these effects often result into the smearing and/or slight altitude displacement of aerosol backscatter features that are seen by both techniques (see Figures 3-4, S3-S4).

As mentioned in Section 4.2.1, the standard deviation of RALMO – COBALD in the boundary layer is strongly influenced by one single outlying profile in our dataset (see Figure 5b). For this reason, it is interesting to note that the interquartile range of the RALMO – COBALD distribution for $z < 2$ km ranges between -20 % and +8 %, and for CHM15K – COBALD between

-22 % and +27 % (see Figures 6f-7f). This highlights the better precision of RALMO measurements with respect to CHM15K in the boundary layer. The slightly higher mean discrepancy between CHM15K and COBALD (+13 %) compared to RALMO (+6 %) could be either due to minor unsolved geometric overlap issues in the ceilometer's retrieval algorithm, or related to the assumption of a constant lidar ratio for all profiles (50 sr) made in the Klett inversion scheme.

For altitudes between 2-6 km, the standard deviations of both RALMO and CHM15K decrease in absolute terms (below 0.13

Mm$^{-1}$ sr$^{-1}$ and 0.16 Mm$^{-1}$ sr$^{-1}$, respectively), while they increase in relative terms (often exceeding 100% of the signal). This is due to the low aerosol content (hence low absolute backscattered signal) in the free troposphere, and the vertically decreasing signal-to-noise ratio of the lidar instruments (especially CHM15K).

Considering the many uncertainties that characterize the retrieval of aerosol backscatter profiles from lidar instruments, from technical and instrumental effects to issues related with the mathematical treatment of the data (e.g., Pappalardo et al., 2004),

our validation using fully independent in-situ measurements is particularly valuable. Our results demonstrate that both single-wavelength (ceilometer) and Raman lidars can provide altitude-resolved measurements that are quantitatively consistent with high-precision balloon-borne measurements over the boundary layer and free troposphere altitude regions. In particular, net of atmospheric variability effects, it is interesting to observe that the discrepancies observed here are comparable to those previ-





ously found between ceilometer and Raman lidars (Matthais et al., 2004) and between different Raman lidar processing algorithms (Pappalardo et al., 2004). Hence, we conclude that aerosol backscatter coefficient measurements by the RALMO and CHM15K lidar systems are in satisfactory agreement with in-situ measurements by COBALD sondes up to 6 km altitude.

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

**Acknowledgements**

This work has been supported by the Swiss National Science Foundation (project no. PZ00P2 168114).

**Data availability**

The RALMO and CHM15K data can be accessed through the EARLINET (www.earlinet.org) and E-PROFILE (www.eumetnet.eu/e-profile) networks, respectively. The COBALD data can be obtained from the authors upon request.





| Technique | Instrument | Type | Light source | Wavelenght(s) | Receiver FOV |
|-----------|-----------|------|--------------|---------------|--------------|
| Remote sensing | RALMO | Raman lidar | Nd:YAG laser | 355 nm | 200 µrad ($\approx 0.01°$) |
| | CHM15K | Ceilometer (elastic backscatter lidar) | Nd:YAG laser | 1064 nm | 450 µrad ($\approx 0.02°$) |
| In-situ | COBALD | Balloon-borne backscatter sonde | LED | 455, 940 nm | 6° |

**Table 1. Summary of the main technical characteristics of the three instruments used in this work, including measuring technique, instrument type, light emitting source, wavelengths and receiver field of view (FOV) angle.**



### RALMO – COBALD (17 profiles, 2016-2019)

| Interval | Before FOV correction | | After FOV correction | |
|---|---|---|---|---|
| | Mean $\Delta\beta_{aer}$(455 nm) | Standard deviation | Mean $\Delta\beta_{aer}$(455 nm) | Standard deviation |
| $0.8 < z < 2$ km a.s.l. | $+ 0.186$ Mm$^{-1}$ sr$^{-1}$ <br> (+ 28.8 %) | $\pm 0.300$ Mm$^{-1}$ sr$^{-1}$ <br> ($\pm$ 45.5 %) | $+ 0.005$ Mm$^{-1}$ sr$^{-1}$ <br> (+ 6.49 %) | $\pm 0.319$ Mm$^{-1}$ sr$^{-1}$ <br> ($\pm$ 40.0 %) |
| $2 < z < 4$ km a.s.l. | $+ 0.010$ Mm$^{-1}$ sr$^{-1}$ <br> (+ 12.2 %) | $\pm 0.101$ Mm$^{-1}$ sr$^{-1}$ <br> ($\pm$ 45.6 %) | $- 0.008$ Mm$^{-1}$ sr$^{-1}$ <br> (+ 6.60 %) | $\pm 0.112$ Mm$^{-1}$ sr$^{-1}$ <br> ($\pm$ 43.3 %) |
| $4 < z < 6$ km a.s.l. | $+ 0.025$ Mm$^{-1}$ sr$^{-1}$ <br> (+ 33.4 %) | $\pm 0.126$ Mm$^{-1}$ sr$^{-1}$ <br> ($\pm$ 95.2 %) | $+ 0.021$ Mm$^{-1}$ sr$^{-1}$ <br> (+ 28.6 %) | $\pm 0.127$ Mm$^{-1}$ sr$^{-1}$ <br> ($\pm$ 95.0 %) |
| $\beta_{aer} > 0.2$ Mm$^{-1}$ sr$^{-1}$ | $+ 0.173$ Mm$^{-1}$ sr$^{-1}$ <br> (+ 25.9 %) | $\pm 0.320$ Mm$^{-1}$ sr$^{-1}$ <br> ($\pm$ 61.4 %) | $- 0.011$ Mm$^{-1}$ sr$^{-1}$ <br> (+ 1.70 %) | $\pm 0.329$ Mm$^{-1}$ sr$^{-1}$ <br> ($\pm$ 56.1 %) |

**Table 2. Overview of the statistical comparison of RALMO vs. COBALD (17 profiles, 2016-2019). For each data interval, we show mean and standard deviation (both in absolute units and percent units relative to COBALD) of the RALMO - COBALD difference in aerosol backscatter coefficient ($\Delta\beta_{aer}$) at 455 nm, calculated both before (left) and after (right) the FOV correction was applied to the COBALD data. Note that the data intervals are the same as those selected for the frequency of occurrence distributions shown in Figure 5 (2 km altitude bins between 0-6 km) and Figure 6 (all data points with COBALD $\beta_{aer} > 0.2$ Mm$^{-1}$ sr$^{-1}$).**





### CHM15K – COBALD (31 profiles, 2014-2019)

| Interval | Before FOV correction | | After FOV correction | |
|---|---|---|---|---|
| | *Mean $\Delta\beta_{aer}$(940 nm)* | *Standard deviation* | *Mean $\Delta\beta_{aer}$(940 nm)* | *Standard deviation* |
| $0.8 < z < 2$ km a.s.l. | + 0.089 Mm$^{-1}$ sr$^{-1}$ (+ 25.2 %) | ± 0.211 Mm$^{-1}$ sr$^{-1}$ (± 53.6 %) | + 0.038 Mm$^{-1}$ sr$^{-1}$ (+ 13.1 %) | ± 0.207 Mm$^{-1}$ sr$^{-1}$ (± 51.0 %) |
| $2 < z < 4$ km a.s.l. | + 0.007 Mm$^{-1}$ sr$^{-1}$ (− 3.71 %) | ± 0.150 Mm$^{-1}$ sr$^{-1}$ (± 322 %) | − 0.013 Mm$^{-1}$ sr$^{-1}$ (− 8.44 %) | ± 0.162 Mm$^{-1}$ sr$^{-1}$ (± 320 %) |
| $4 < z < 6$ km a.s.l. | − 0.015 Mm$^{-1}$ sr$^{-1}$ (− 21.7 %) | ± 0.119 Mm$^{-1}$ sr$^{-1}$ (± 644 %) | − 0.020 Mm$^{-1}$ sr$^{-1}$ (− 23.5 %) | ± 0.139 Mm$^{-1}$ sr$^{-1}$ (± 641 %) |
| $\beta_{aer} > 0.1$ Mm$^{-1}$ sr$^{-1}$ | + 0.083 Mm$^{-1}$ sr$^{-1}$ (+ 16.2 %) | ± 0.275 Mm$^{-1}$ sr$^{-1}$ (± 43.2 %) | − 0.001 Mm$^{-1}$ sr$^{-1}$ (+ 1.93  %) | ± 0.281 Mm$^{-1}$ sr$^{-1}$ (± 42.5 %) |

**Table 3. Overview of the statistical comparison of CHM15K vs. COBALD (31 profiles, 2014-2019). For each data interval, we show mean and standard deviation (both in absolute units and percent units relative to COBALD) of the CHM15K - COBALD difference in aerosol backscatter coefficient ($\Delta\beta_{aer}$) at 940 nm, calculated both before (left) and after (right) the FOV correction was applied to the COBALD data. Note that the data intervals are the same as those selected for the frequency of occurrence distributions shown in Figure 8 (2 km altitude bins between 0-6 km) and Figure 9 (all data points with COBALD $\beta_{aer} > 0.1$ Mm$^{-1}$ sr$^{-1}$).**

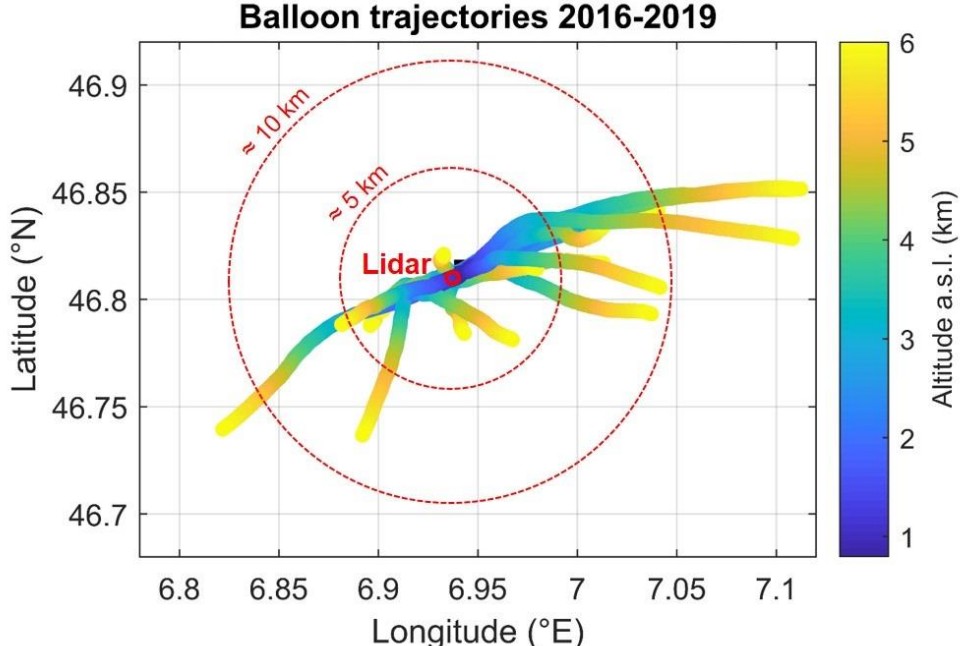

**Figure 1. Overview of balloon trajectories as function of latitude, longitude and altitude (color scale), for all the analyzed soundings in the period 2016-2019 (total 17 profiles, corresponding to the dataset used for the RALMO vs. COBALD comparison). The balloon trajectories are color-coded with altitude and plotted with vertical resolution of 30 m between 800 m - 6 km altitude asl. The location of the RALMO and CHM15K lidars (and balloon launching site) is shown by the solid red circle (46.82°E, 6.95°N). Two dotted red circles indicate horizontal distances of approximately 5 km and 10 km from the lidar site.**





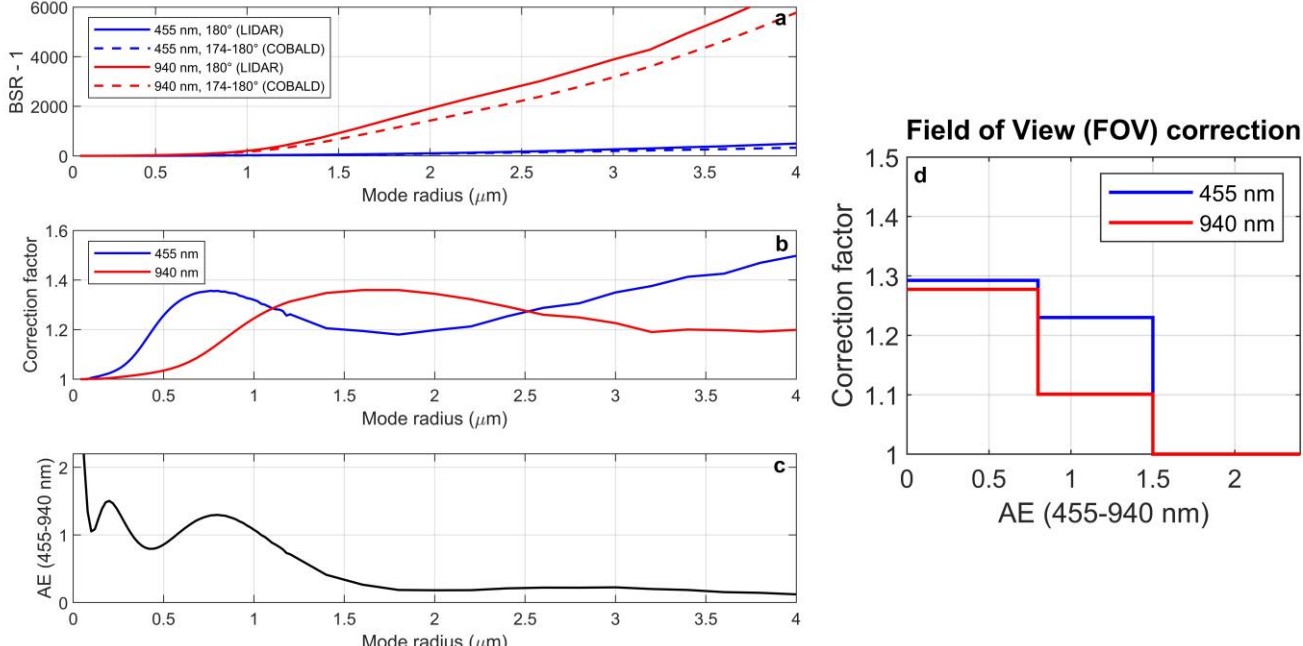

**Figure 2.** Overview of Mie-theory scattering model simulations. Panel (a): backscatter ratio (BSR) – 1 as function of mode radius ($R_m$) at 455 nm (blue) and 940 nm (red) calculated assuming a field of view (FOV) angle of 174°-180° (dashed lines, COBALD) and 180° (solid lines, lidar). Panel (b): correction factors, i.e. lidar-to-COBALD ratio of BSR – 1 (as shown in Panel a) for 455 nm (blue) and 940 nm (red), as function of $R_m$. Panel (c): simulated Angstrom exponent (AE) for the COBALD wavelength interval (455-940 nm), as function of $R_m$. Panel (d): resulting FOV correction as function of AE (see discussion in Section 3.3).



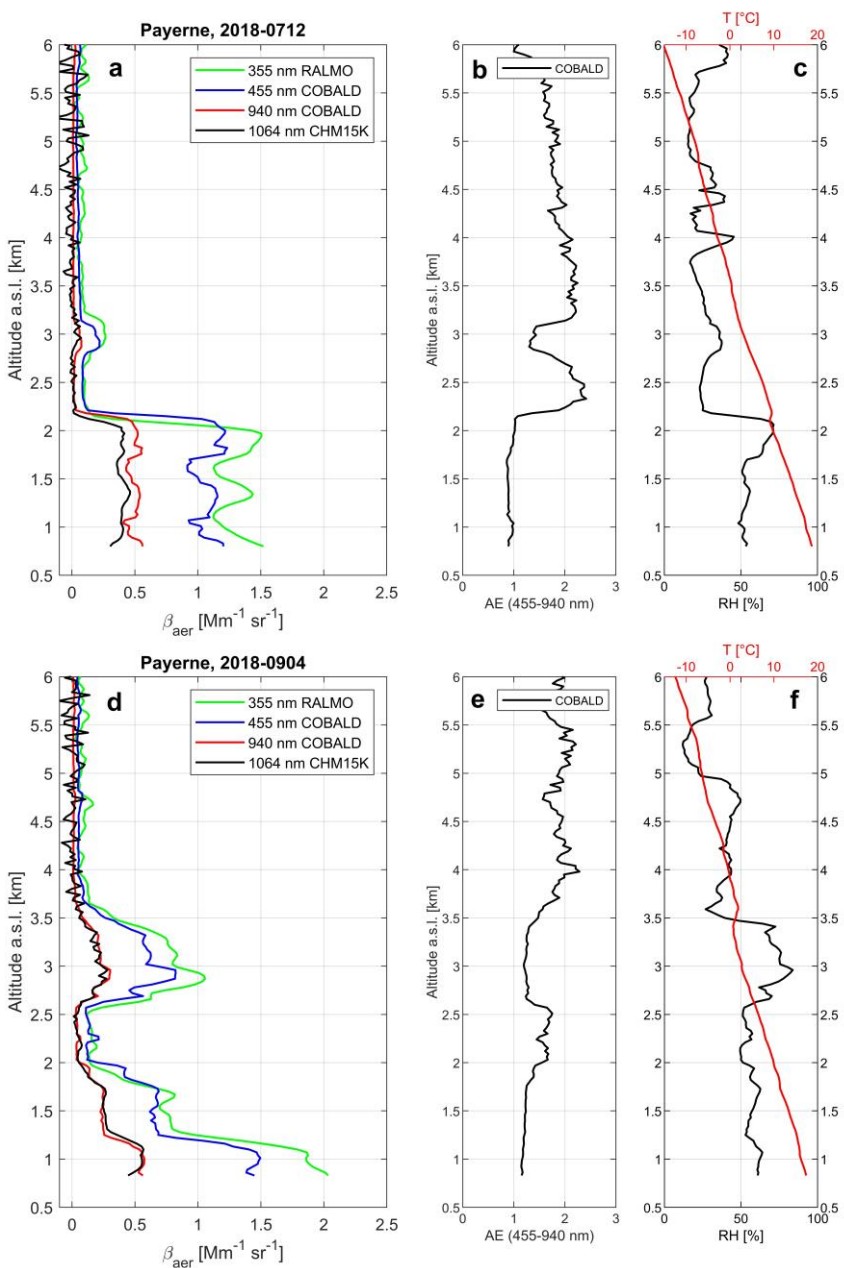

**Figure 3. Overview of selected profiles measured on 12 July 2018 (Panels a-c) and 4 September 2018 (Panels d-f). Panels (a, c): vertical profiles of aerosol backscatter coefficient ($\beta_{aer}$) as function of altitude, measured by RALMO (355 nm, green), COBALD (455 nm, blue and 940 nm, red) and CHM15K (1064 nm, black). Panels (b, d): vertical profiles of Angstrom exponent (AE) for wavelengths 455-940 nm, calculated from the COBALD data. Panels (c, f): vertical profiles of relative humidity (RH, black) and temperature (red, top scale) measured by the Vaisala RS41-SGP radiosonde (flying in tandem with the COBALD sonde).**


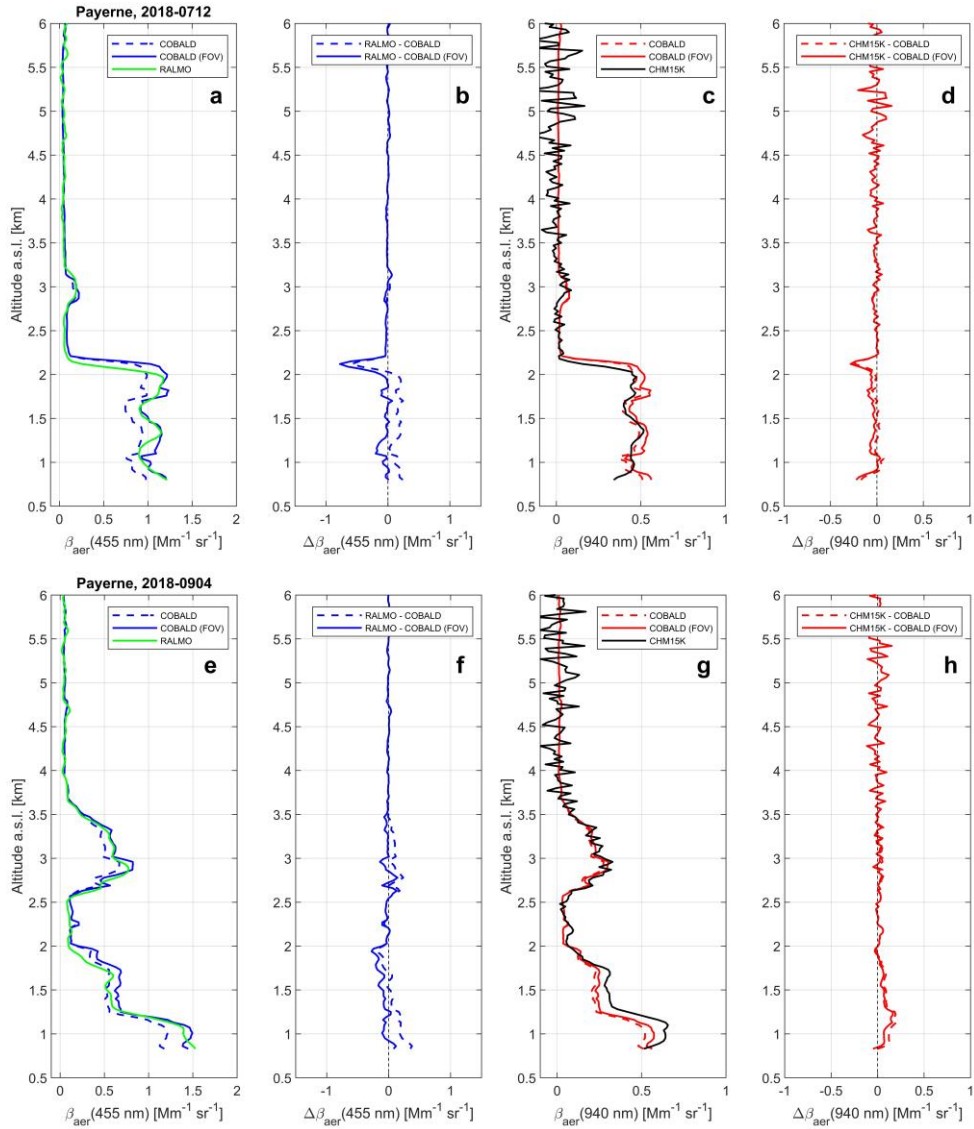

**Figure 4. Quantitative comparison of RALMO vs. COBALD (Panels a-b, e-f) and CHM15K vs. COBALD (Panels c-d, g-h) for the selected profiles measured on 12 July 2018 (Panels a-d) and 4 September 2018 (Panels e-h). Panels (a, e): vertical profiles of aerosol backscatter coefficient ($\beta_{aer}$) at 455 nm measured by RALMO (green) and COBALD (blue), both without (dashed) and with (solid) application of the FOV correction. Panels (b, f): vertical profiles of the RALMO – COBALD difference in $\beta_{aer}$ ($\Delta\beta_{aer}$) at 455 nm, both without (dashed) and with (solid) application of the FOV correction. Panels (c, g): vertical profiles of $\beta_{aer}$ at 940 nm measured by CHM15K (black) and COBALD (red), both without (dashed) and with (solid) FOV correction. Panels (d, h): vertical profiles of $\Delta\beta_{aer}$ for CHM15k – COBALD at 940 nm, both without (dashed) and with (solid) FOV correction.**



**Figure 5. Statistical comparison of RALMO vs. COBALD (17 profiles, 2016-2019): vertical profiles. Panel (a): all data points (blue circles), mean profile (thick solid black line) and mean ± standard deviation profiles (thin dashed black lines) of RALMO – COBALD difference in aerosol backscatter coefficient ($\Delta\beta_{aer}$) at 455 nm, as function of altitude. Panel (b): same as Panel (a), with $\Delta\beta_{aer}$ expressed in percent units (%) relative to the COBALD measurements (instead of absolute backscatter coefficient units, Mm$^{-1}$ sr$^{-1}$).**

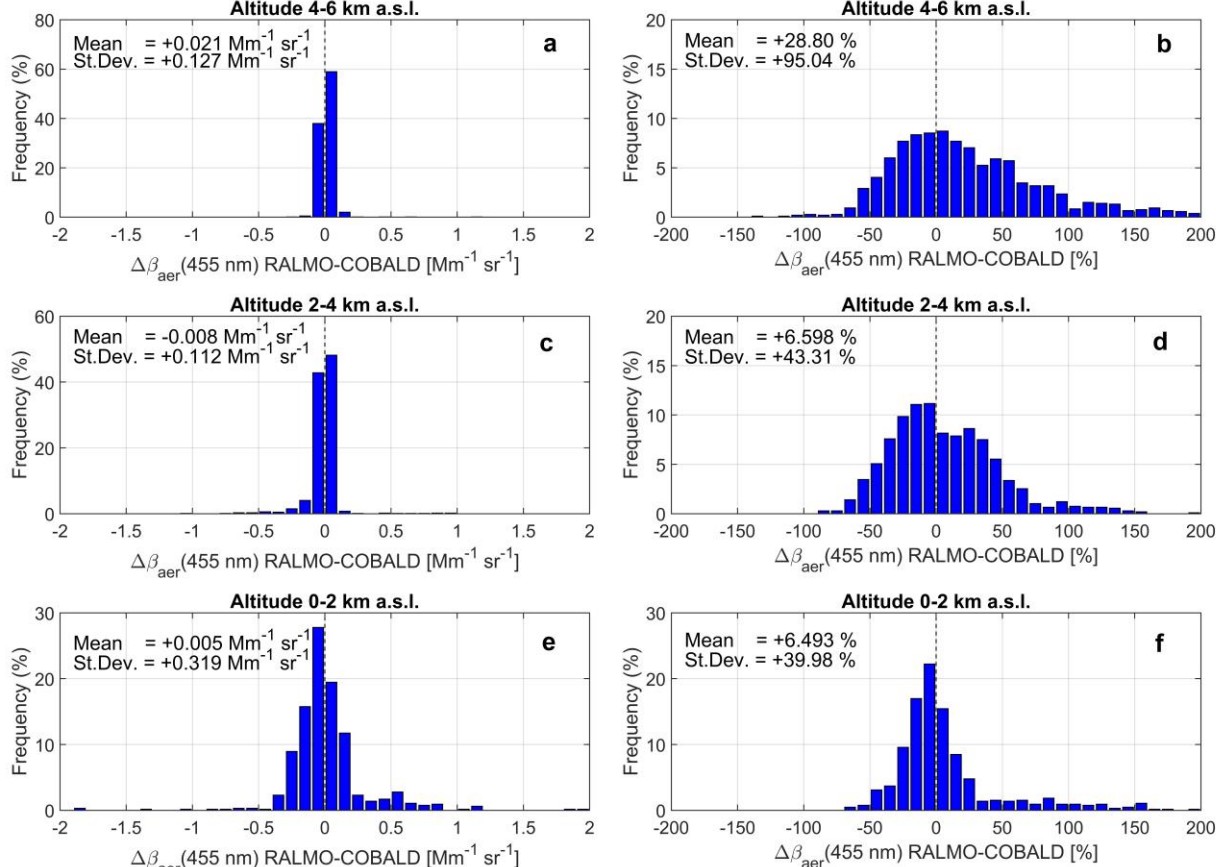

**Figure 6. Statistical comparison of RALMO vs. COBALD (17 profiles, 2016-2019): frequency of occurrence distributions. Panels (a, c, e): frequency of occurrence distributions of RALMO – COBALD difference in aerosol backscatter coefficient ($\Delta\beta_{aer}$) at 455 nm, for the altitude intervals 0-2 km (Panel e), 2-4 km (c) and 4-6 km (a) above sea level (a.s.l.). Panels (b, d, f): same as Panels (a, c, e), with $\Delta\beta_{aer}$ expressed in percent units (%) relative to the COBALD measurements (instead of absolute backscatter coefficient units, Mm$^{-1}$ sr$^{-1}$). Mean value and standard deviation of the distributions are displayed in each panel. The frequency of occurrence distributions are calculated in $\Delta\beta_{aer}$ intervals of 0.1 Mm$^{-1}$ sr$^{-1}$ (Panels a, c, e) and 10% (Panels a, c, e).**

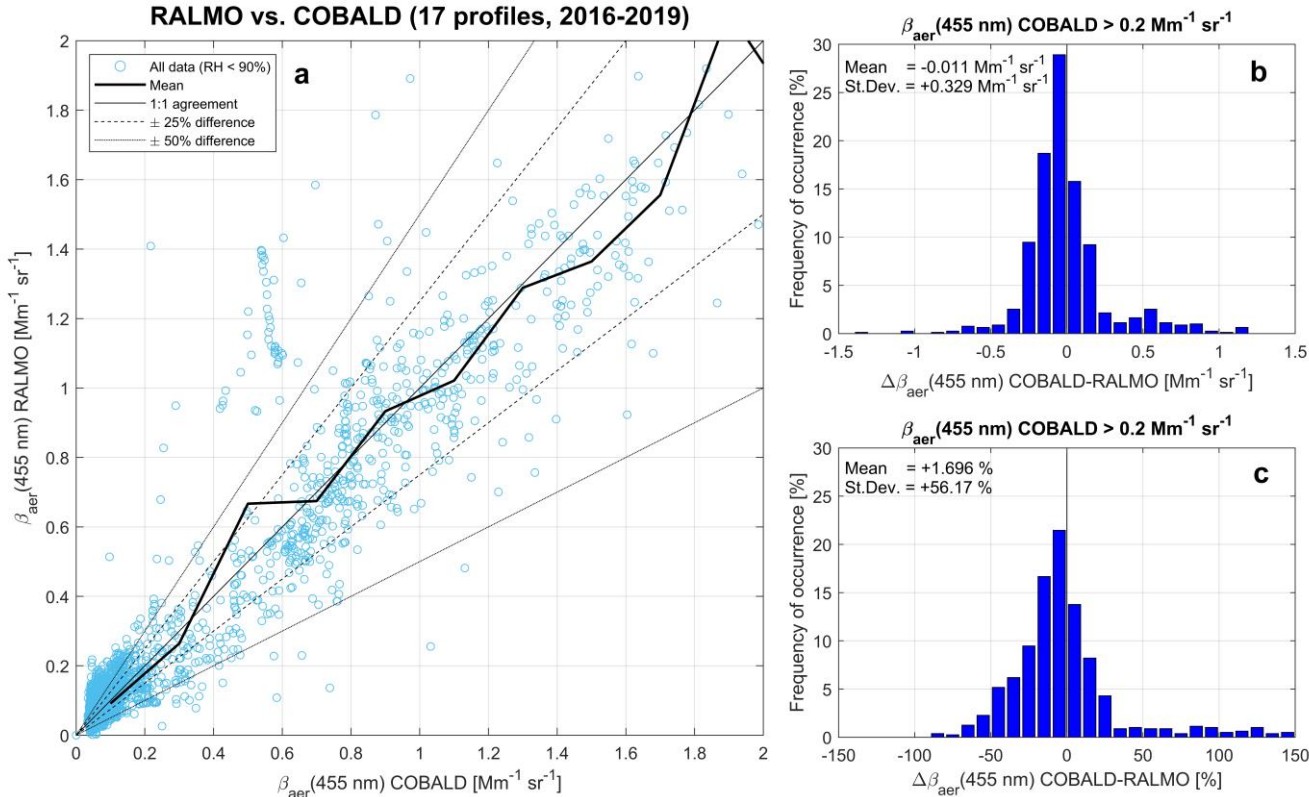

**Figure 7. Statistical comparison of RALMO vs. COBALD (17 profiles, 2016-2019): scatter plot and frequency of occurrence distributions. Panel (a): scatter plot of all data points (blue circles) of aerosol backscatter coefficient ($\beta_{aer}$) at 455 nm measured by RALMO (y-axis) vs. $\beta_{aer}$ at 455 nm measured by COBALD (x-axis). Thin black lines identify the 1:1 agreement isoline between RALMO and COBALD (solid), ±25 % differences (dashed) and ±50 % differences (dotted). The thick solid black lines shows the mean $\beta_{aer}$ from RALMO calculated for each 0.2 Mm$^{-1}$ sr$^{-1}$ interval of $\beta_{aer}$ from COBALD. Panels (b-c): frequency of occurrence distributions of RALMO – COBALD difference in $\beta_{aer}$ ($\Delta\beta_{aer}$), calculated for all data points with COBALD $\beta_{aer}^{455}$ > 0.2 Mm$^{-1}$ sr$^{-1}$ at 455 nm.**

**Figure 8. Statistical comparison of CHM15K vs. COBALD (31 profiles, 2014-2019): vertical profiles. Panel (a): all data points (blue circles), mean profile (thick solid black line) and mean ± standard deviation profiles (thin dashed black lines) of CHM15K – COBALD difference in aerosol backscatter coefficient ($\Delta\beta_{aer}$) at 940 nm, as function of altitude. Panel (b): same as Panel (a), with $\Delta\beta_{aer}$ expressed in percent units (%) relative to the COBALD measurements (instead of absolute backscatter coefficient units, Mm$^{-1}$ sr$^{-1}$).**

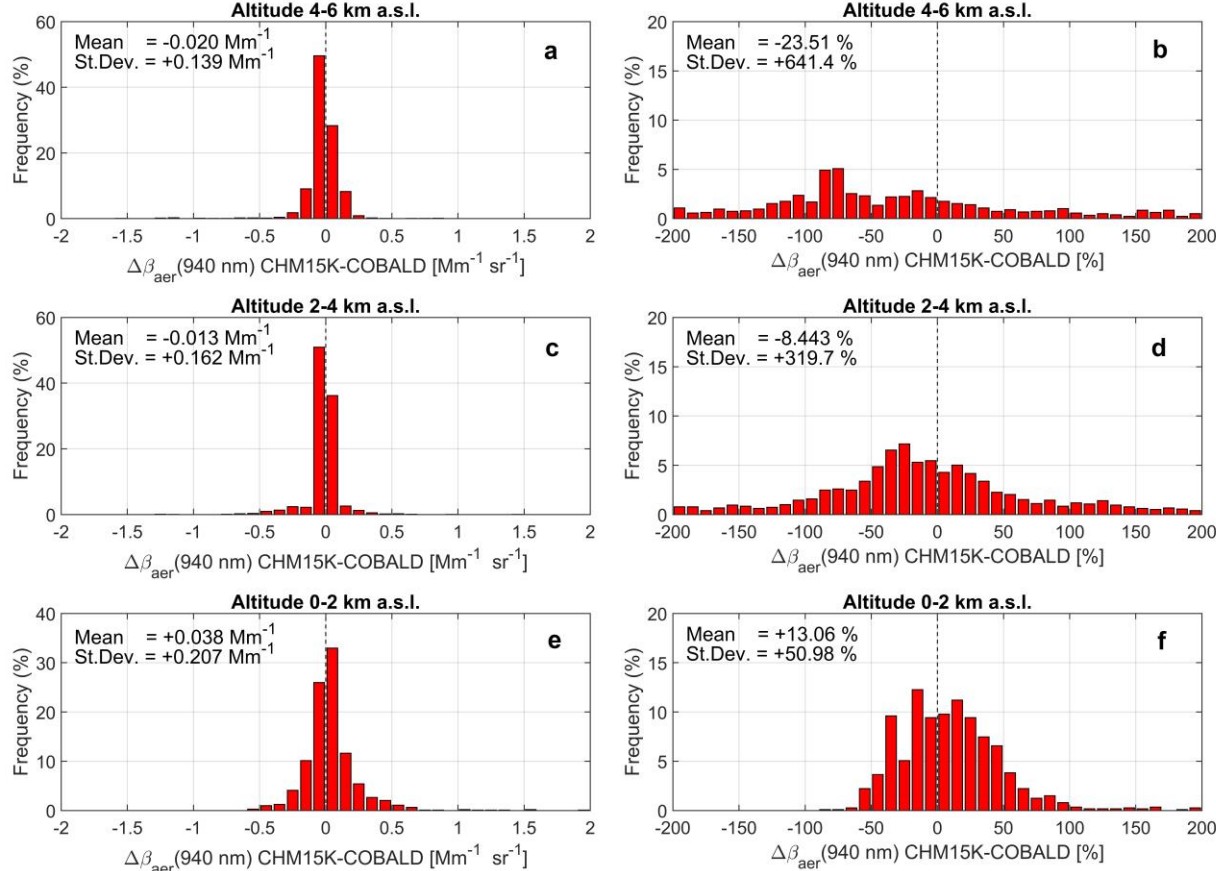

**Figure 9. Statistical comparison of CHM15K vs. COBALD (31 profiles, 2014-2019): frequency of occurrence distributions. Panels (a, c, e): frequency of occurrence distributions of CHM15K – COBALD difference in aerosol backscatter coefficient ($\Delta\beta_{aer}$) at 940 nm, for the altitude intervals 0-2 km (Panel e), 2-4 km (c) and 4-6 km (a) above sea level (a.s.l.). Panels (b, d, f): same as Panels (a, c, e), with $\Delta\beta_{aer}$ expressed in percent units (%) relative to the COBALD measurements (instead of absolute backscatter coefficient units, Mm$^{-1}$ sr$^{-1}$). Mean value and standard deviation of the distributions are displayed in each panel. The frequency of occurrence distributions are calculated in $\Delta\beta_{aer}$ intervals of 0.1 Mm$^{-1}$ sr$^{-1}$ (Panels a, c, e) and 10% (Panels a, c, e).**



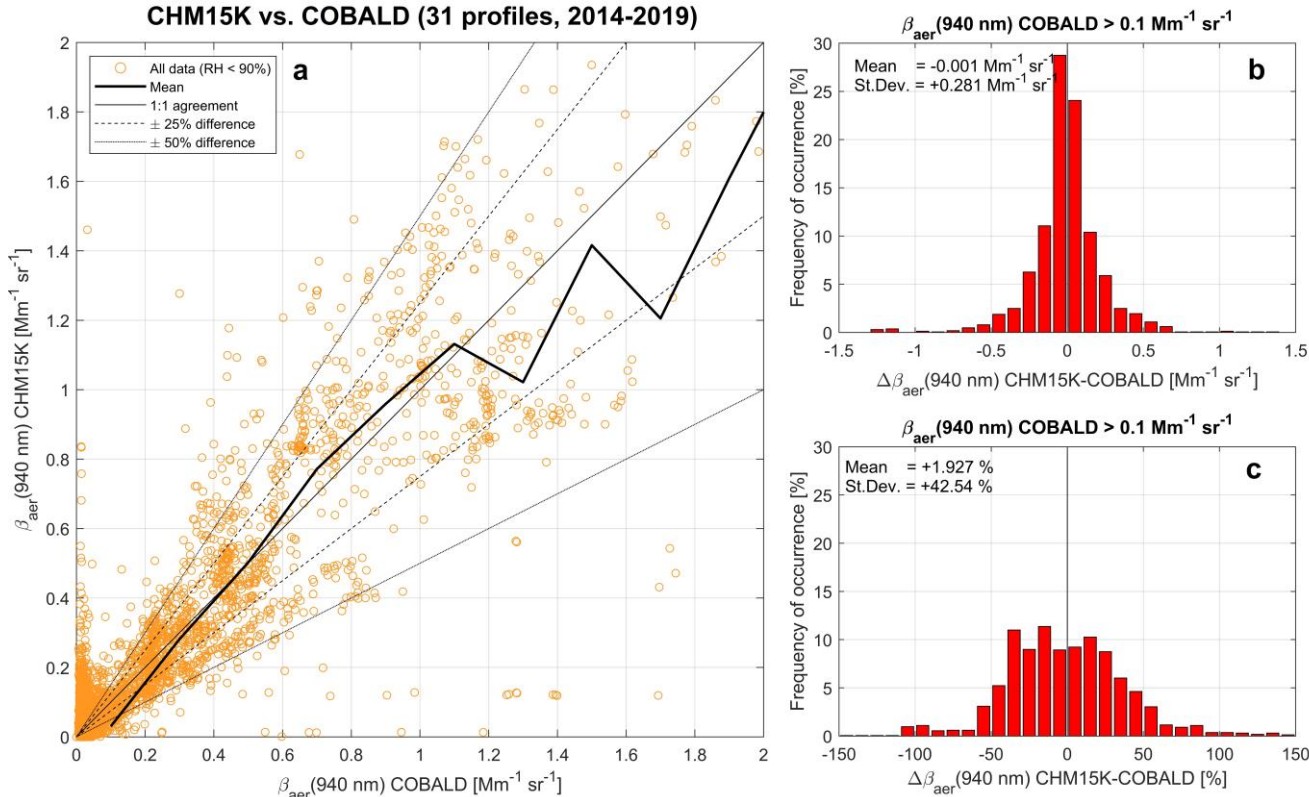

**Figure 10. Statistical comparison of CHM15K vs. COBALD (31 profiles, 2014-2019): scatter plot and frequency of occurrence distributions. Panel (a): scatter plot of all data points (blue circles) of aerosol backscatter coefficient ($\beta_{aer}$) at 940 nm measured by CHM-15K (y-axis) vs. $\beta_{aer}$ at 455 nm measured by COBALD (x-axis). Thin black lines identify the 1:1 agreement isoline between CHM15K and COBALD (solid), ±25 % differences (dashed) and ±50 % differences (dotted). The thick solid black lines shows the mean $\beta_{aer}$ from CHM15K calculated for each 0.2 Mm$^{-1}$ sr$^{-1}$ interval of $\beta_{aer}$ from COBALD. Panels (b-c): frequency of occurrence distributions of CHM15K – COBALD difference in $\beta_{aer}$ ($\Delta\beta_{aer}$), calculated for all data points with COBALD $\beta_{aer}^{940}$ > 0.1 Mm$^{-1}$ sr$^{-1}$ at 940 nm.**