# Peer review of "Validation of aerosol backscatter profiles from Raman lidar and ceilometer using balloon-borne measurements"

_Atmospheric Chemistry and Physics, 2020_

## Referee Comment (RC1) · Anonymous Referee #1 · 1 Jul 2020

The paper presents an interesting comparison between the aerosol backscatter coefficients measured by two different lidar systems (a sophisticated multiwavelength lidar with elastic and Raman channels, and a ceilometer) and those obtained by a balloon-borne instrument performing in-situ measurements. The latter is taken as reference to validate the backscatter profiles provided by the lidars.

The paper is well written and describes a sound methodology that, besides providing the validation mentioned in the paper's title, can be useful for similar verifications at other sites.

I think the paper is worth publishing (although given its scope, focusing on techniques and methods rather than on atmospheric processes, perhaps the sister journal Atmospheric Measurement Techniques would provide a more suitable forum).

The authors may wish to consider the following remarks that in my view would improve the manuscript.

**Main remarks**

1. In the paper it is implied that the COBALD instrument is taken as the reference against which the lidar-derived backscatter coefficients are validated, on grounds that an in-situ instrument inherently provides "higher precision and signal-to-noise ratio compared to remote sensing measurements" (line 31, page 2). For this reason, I miss a more detailed description of the instrument specifications, namely, systematic (bias) and random (noise) error.

2. The above remark is somewhat linked to a seemingly lack of explanation for the mean deviations between the lidar-derived backscatter coefficients and those provided by the COBALD instruments in the PBL (+ 6% for RALMO and +13% for the ceilometer, below 2 km, (lines 23-24, page 1)). Is this just a random effect resulting from the limited dataset? Might negative differences be obtained for other datasets? Is this an effect resulting from the wavelength conversion and the FOV correction discussed respectively in sections 3.2 and 3.3? Does it come from other reasons (see next point).

3. Related to the previous point, the authors put forward the possibility (lines 1-2 of page 13) that the 15% positive bias below 2.5 km (line 30 of page 12; by the way, shouldn't it be rather 13%, cf. line 24 of page 1 and fig. 9f) be related to "minor unsolved geometric overlap issues in the ceilometer's retrieval algorithm, or (more likely) related to the assumption of a constant lidar ratio (50 sr)". Could this also be the cause for the (smaller (+6.5%)) positive bias in the RALMO vs. COBALD comparison below 2 km? The influence of an assumed lidar ratio could checked with relative ease. Have the authors done it?

4. It would also be advisable that the authors provide some indication on the statistical error in the measurements (estimated error bars), not only for the COBALD sondes, but also for the lidar-derived backscatter coefficients. That would help clarifying how much of the standard deviation found in the comparisons presented is due to the uncertainty of the measurements of each instrument, which must set a lower limit to that standard deviation affected as well, as the authors point out, by the differences between the atmosphere volumes measured by the sonde and by the lidars.

5. I would suggest restricting the use of relative differences in the comparisons of the backscatter coefficients to the layers with a medium to high aerosol content. I think using it in zones of low aerosol content or in the free troposphere is misleading, as small absolute differences will yield large figures when they are divided by a very small backscatter coefficient, which in turn is probably driven by statistical noise. In this respect, the authors should probably follow the criteria stated in section 4D of their reference Matthais et al., 2004. If the authors want to highlight something important coming out from these comparisons in terms of relative errors at those altitude ranges with little or no aerosol content, they should be more explicit.

**Minor issues**

1. The statistical analyses of the comparisons of RALMO and the CHM15K ceilometer against COBALD are divided in figs. 6 and 9 in altitude zones, the first one being 0-2 km asl. However, in the text, when discussing the comparisons at the lowermost altitudes, the authors often use the 2.5 km limit (e.g. lines 25 and 28 in page 10, etc.). It would be easier for the reader to follow the discussions if the text and the figures would use the same limits.

2. Page 2, line 26: "the atmospheric number density". I would suggest "the atmospheric number density of molecules". Note that the pressure-to-temperature ratio would also do.

3. While the minimum height of measurements for the CHM15K instrument is indirectly given through its full overlap range, this information seems to be missing for the RALMO system. Even though the Raman technique employed in RALMO to derive the aerosol backscatter coefficient allows compensating incomplete overlap effects to some extent, I think RALMO's minimum usable altitude should be stated for completeness.

4. Note a possible inconsistence in the full overlap range of CHM15K. In line 4 of page 4 it is stated as "800 m above the station", while in lines 2-3 of page 6 is it is said that "We only select measurements from  $\approx$  300 m above the ground station in order to minimize the effect a possible incomplete overlap of the lidar systems in the lower part of the profiles". Left aside the already mentioned fact that no overlap information seems to be given for RALMO, do the authors use CHM15K data obtained below its full overlap range? This deserves some calrification.

5. I suggest that a logarithmic scale be used for the horizontal axes in figs. 2a, b, and c.

6. In the caption of fig. 2 it should be stated that the graph in panel a is obtained for  $N=10^3$  cm-3. Currently that information is found only in the main text.

7. I suggest trying to find a symbol (and give a name) for the ratio  $\beta_{aer} / \beta_{mol}$ . Otherwise the authors have to use the rather awkward notation BSR-1 to refer to that ratio and the text may even fall in ambiguities, for example in line 3 of page 8 when they say "Figure 2a shows the

simulated aerosol backscatter ratio (i.e. BSR – 1)". But the backscatter ratio is BSR, not BSR-1. Perhaps just  $\beta_{aer} / \beta_{mol}$  would do.

---

## Referee Comment (RC2) · Anonymous Referee #2 · 7 Jul 2020

The manuscript shows an intercomparison between a Raman lidar, a commercial ceilometer and an optical in-situ sonde takes as reference. This contribution doesn't represent a substantial contribution to scientific progress and I agree that it is more indicated for Atmospheric Measurement Techniques than ACP.

Major Issues:

The FOV correction factor has been computed with a well defined aerosol distribution. How the results change if a more likely bi-modal distribution is used instead? Or changing the distribution width and/or the refractive index? As it is implemented, the correction is depending on a particular type of aerosol.

[Figure]

The statistical intercomparison is unclear and counter-intuitive. I would suggest to the authors to use the Pearson Cross Correlation coefficient paired with the Root Mean Square Error, on the whole atmospheric profile and at different altitude ranges, e.g. into the PBL, free troposphere... Figure 5 and 8 are unnecessary.

CHM15K inversion should be explained more in detail, as the molecular signal can be very low at 1064nm.

Optical measurements are strongly affected by water vapor absorption at 940nm. Usually, the ceilometers at this wavelength use a radiative transfer computation to correct the profile. What about COBALD sonde? In the text it is not mentioned. Moreover, some equations are needed to better explain lines 25-28 (Pag. 4).

It would be more interesting to intercompare the two instruments vs. COBALD for different meteorological conditions and aerosol loading.

---

## Author Response (AR1)

**Manuscript acp-2020-294 – Author's replies to reviewers**

We thank gratefully the editor and two anonymous referees for careful reading and comments. Below are the referee's comments in **black**, and replies from the authors in **blue**. Please note that page and line numbers given below refer to the revised version of the manuscript without tracked-changes.

**Anonymous Referee #1**

The paper presents an interesting comparison between the aerosol backscatter coefficients measured by two different lidar systems (a sophisticated multiwavelength lidar with elastic and Raman channels, and a ceilometer) and those obtained by a balloon-borne instrument performing in-situ measurements. The latter is taken as reference to validate the backscatter profiles provided by the lidars.

The paper is well written and describes a sound methodology that, besides providing the validation mentioned in the paper's title, can be useful for similar verifications at other sites.

I think the paper is worth publishing (although given its scope, focusing on techniques and methods rather than on atmospheric processes, perhaps the sister journal Atmospheric Measurement Techniques would provide a more suitable forum).

The authors may wish to consider the following remarks that in my view would improve the manuscript.

**Main remarks**

1. In the paper it is implied that the COBALD instrument is taken as the reference against which the lidar-derived backscatter coefficients are validated, on grounds that an in-situ instrument inherently provides "higher precision and signal-to-noise ratio compared to remote sensing measurements" (line 31, page 2). For this reason, I miss a more detailed description of the instrument specifications, namely, systematic (bias) and random (noise) error.

The accuracy and precision of COBALD BSR were estimated by Vernier et al. (2015) (Section 2.1) as 5 % and 1 %, respectively, at upper tropospheric conditions (which can be regarded as an upper limit here, due to the higher absolute signal measured in the lower troposphere). This previously missing information is now included in the manuscript (Page 5, Lines 7-8).

2. The above remark is somewhat linked to a seemingly lack of explanation for the mean deviations between the lidar-derived backscatter coefficients and those provided by the CO-BALD instruments in the PBL (+ 6% for RALMO and +13% for the ceilometer, below 2 km, (lines 23-24, page 1)). Is this just a random effect resulting from the limited dataset? Might negative differences be obtained for other datasets? Is this an effect resulting from the wavelength conversion and the FOV correction discussed respectively in sections 3.2 and 3.3? Does it come from other reasons (see next point).

Indeed, the uncertainties related with the wavelength conversion and the FOV correction, as well as spatial and temporal variability effects, contribute to the mean and standard deviations of our statistical comparison. A more detailed and quantitative discussion of the uncertainties associated with these effects is now included in the new Section 4.3.

In particular, concerning the wavelength conversion: "From Equation 1 we can derive that an error of 0.2 in AE, which is a conservative estimate considering the small difference between the wavelengths that are compared, results in an error of 5 % in  $\beta_{aer}$  for the 355-to-455 nm conversion, and 2.5 % for the 1064-to-940 nm conversion" (Page 15, Lines 9-11).

Concerning the FOV correction: "From Figure 2b, we can estimate an uncertainty of up to  $\pm$  20% in  $\beta$ aer for the PBL (for both 455 and 940 nm), due to variability of the correction factors in the range of AE = 0.8-1.5, which is not resolved by the parameterization of the FOV correction factors in AE-space." (Page 15, Lines 12-14).

Finally, about spatial and temporal variability: "These effects can lead to large discrepancies over small altitude layers, as in the case of strong vertical gradients in  $\beta$ aer (e.g., top of boundary layer: Figure 4a-d), as well as potentially over larger altitude regions, due to the horizontal gradient of the  $\beta$ aer field around the station (e.g., in the case of the strongly outlying profiles of the statistical comparison: see Figures 5, 8). The lack of information on the aerosol size distribution and the high spatial and temporal variability of atmospheric aerosols prevent an accurate quantification of these artifacts, which inevitably affect the standard deviations of our statistical comparison" (Page 15, Lines 16-21).

Hence, we conclude: "Despite of these limitations, the comparison of individual profiles (Section 4.1) shows that both RALMO and CHM15K are able to achieve an excellent agreement with COBALD measurements, including the correct representation of fine and complex structures in the  $\beta$ aer vertical profiles (Figures 3-4, S2-S3). In particular, the case study of 12 July 2018 (Figures 4a-d) shows differences between the lidars and COBALD which are smaller than expected statistical uncertainty associated with the remote sensing measurements alone (10-15 %, see Section 2.1). This suggests that, under optimal conditions (such as, no wind shear, uniform  $\beta$ aer field, mono-modal aerosol size distribution), the deviations between the two lidars and COBALD are typically smaller than the average  $\sigma$  of our statistical comparison." (Page 15, Lines 28 - Page 16, Line 1).

3. Related to the previous point, the authors put forward the possibility (lines 1-2 of page 13) that the 15% positive bias below 2.5 km (line 30 of page 12; by the way, shouldn't it be rather 13%, cf. line 24 of page 1 and fig. 9f) be related to "minor unsolved geometric overlap issues in the ceilometer's retrieval algorithm, or (more likely) related to the assumption of a constant lidar ratio (50 sr)". Could this also be the cause for the (smaller (+6.5%)) positive bias in the RALMO vs. COBALD comparison below 2 km? The influence of an assumed lidar ratio could checked with relative ease. Have the authors done it?

The uncertainty associated with the assumption of a constant lidar ratio (50 sr) in the retrieval algorithm for CHM15K, which adds up to the uncertainties discussed in the previous comment, is now also discussed more in detail in Section 4.3.

"Using a similar ceilometer (Jenoptik CHM15kx), Wiegner and Geiss (2012) estimate that an error of  $\pm$  10 sr in lidar ratio leads to an error in  $\beta$ aer smaller than 2 % in the boundary layer. Ackermann (1998) shows that 50  $\pm$  10 sr represents well the expected range of variability of the lidar ratio of continental aerosol in the infrared spectrum, for all RH conditions between 0-90 %. Therefore, this uncertainty conceivably plays a minor role compared to the effects discussed above." (Page 15, Lines 24-27).

For RALMO, no a-priori assumption on lidar ratio is required for the backscatter retrieval (see Section 2.1). Therefore, this uncertainty only affects the CHM15K comparison. The sentence at Page 13, Lines 1-2 of the original manuscript was removed within the revision of the statistical comparison (see answer to comment 5 below).

4. It would also be advisable that the authors provide some indication on the statistical error in the measurements (estimated error bars), not only for the COBALD sondes, but also for the lidar-derived backscatter coefficients. That would help clarifying how much of the standard deviation found in the comparisons presented is due to the uncertainty of the measurements of each instrument, which must set a lower limit to that standard deviation affected as well, as the authors point out, by the differences between the atmosphere volumes measured by the sonde and by the lidars.

Information on the estimated mean statistical uncertainty of RALMO and CHM15K is now provided in Section 2.1. For RALMO "The mean statistical uncertainties associated with the retrieval of  $\beta$ aer at 355 nm from Raman inversion techniques are typically estimated as 15 % in the PBL (Pappalardo et al., 2004)." (Page 4, Lines 2-4). For CHM15K: "using a similar ceilometer (CHM15kx by Jenoptik, Germany), Wiegner and Geiss (2012) report a relative error of 10 % on  $\beta$ aer at 1064 nm retrieved by this method" (Page 4, Lines 14-15). The observed standard

deviations of the statistical comparison, as well as the comparison of individual profiles, are now discussed in the context of these statistical uncertainties in Section 4.3 and Conclusions (see in particular Page 15 Line 30 - Page 16 Line 1, Page 17 Lines 4-7).

5. I would suggest restricting the use of relative differences in the comparisons of the backscatter coefficients to the layers with a medium to high aerosol content. I think using it in zones of low aerosol content or in the free troposphere is misleading, as small absolute differences will yield large figures when they are divided by a very small backscatter coefficient, which in turn is probably driven by statistical noise. In this respect, the authors should probably follow the criteria stated in section 4D of their reference Matthais et al., 2004. If the authors want to highlight something important coming out from these comparisons in terms of relative errors at those altitude ranges with little or no aerosol content, they should be more explicit.

Following this comment and one remark of Reviewer #2, the statistical comparison (including mainly Section 4 and Figures 5-10) has been strongly revised and improved. In particular, the following changes were made:

- Low and medium-high aerosol content measurements are separated according to an empirical threshold, and shown by different colors in Figures 5, 7, 8, 10.
- Mean and standard deviations profiles (Figures 5, 8) are calculated from medium-high aerosol content only, and the use of relative differences in the discussion is mainly restricted to medium-high aerosol content measurements.
- The probability density functions (Figures 6, 9) are calculated for medium-high aerosol content and all data separately.
- The Pearson correlation coefficient is now also calculated for medium-high aerosol content data, allowing to quantitatively evaluate the linearity of the correlation between the lidar and COBALD measurements.
- A new Section 3.3 (Page 9 Line 10 Page 10 Line 17) was added to the manuscript, where we introduce the sorting of the data according to aerosol content, and formally define the compared quantities (Equations 3-6).

This approach allows to meaningfully quantify the deviations of medium-high aerosol content measurements at all altitudes, which is the main focus of this study, while at the same time not fully neglecting low aerosol content measurement at higher altitudes, which are important because a good agreement in the free troposphere ensures that all profiles are well calibrated (see Page 9, Lines 24-26).

We believe the revised statistical comparison is strongly improved compared to the previous version of the paper, both in terms of scientific content and clarity, and we thank the reviewer for this comment.

**Minor issues**

1. The statistical analyses of the comparisons of RALMO and the CHM15K ceilometer against COBALD are divided in figs. 6 and 9 in altitude zones, the first one being 0-2 km asl. However, in the text, when discussing the comparisons at the lowermost altitudes, the authors often use the 2.5 km limit (e.g. lines 25 and 28 in page 10, etc.). It would be easier for the reader to follow the discussions if the text and the figures would use the same limits.

Done. In the revised statistical comparison, we reduced the number of vertical intervals from three (0.8-2 km, 2-4 km, 4-6 km) to two (0.8-3 km, 3-6 km), and the limit value of z = 3 km is now used consistently throughout the manuscript.

2. Page 2, line 26: "the atmospheric number density". I would suggest "the atmospheric number density of molecules". Note that the pressure-to-temperature ratio would also do.

**Done (page 2 line 26).**

3. While the minimum height of measurements for the CHM15K instrument is indirectly given through its full overlap range, this information seems to be missing for the RALMO system. Even though the Raman technique employed in RALMO to derive the aerosol backscatter coefficient allows compensating incomplete overlap effects to some extent, I think RALMO's minimum usable altitude should be stated for completeness.

Indeed, thanks to its Raman retrieval technique, the RALMO backscatter is unaffected by incomplete overlap issues. Nevertheless, the signal-to-noise is typically very low in the first 200 m above the station, hence this can be considered as a 'minimum usable altitude'. This information is now included in the manuscript (Page 3, Lines 28-30).

4. Note a possible inconsistence in the full overlap range of CHM15K. In line 4 of page 4 it is stated as "800 m above the station", while in lines 2-3 of page 6 is it is said that "We only select measurements from  $\approx$  300 m above the ground station in order to minimize the effect a possible incomplete overlap of the lidar systems in the lower part of the profiles". Left aside the already mentioned fact that no overlap information seems to be given for RALMO, do the authors use CHM15K data obtained below its full overlap range? This deserves some calrification.

Below the full overlap altitude of CHM15K (800 m above station), the backscatter profiles are corrected for the incomplete overlap between the incoming beam and the receiver's field of view, as described in Hervo et al. (2016). This (previously missing) information is now included in the manuscript (Page 4, Line 9).

The rejection of all measurements below 300 m above the station is aimed to avoid the region of maximum incomplete overlap of CHM15K, as well as to avoid the region of low signal-tonoise ratio of RALMO at low altitudes (see answer to previous comment). This statement is now clarified in the manuscript (Page 6, Lines 23-25).

5. I suggest that a logarithmic scale be used for the horizontal axes in figs. 2a, b, and c. 6. In the caption of fig. 2 it should be stated that the graph in panel a is obtained for 3 -3 N=10 cm .Currently that information is found only in the main text. 7. I suggest trying to find a symbol (and give a name) for the ratio /  $\beta$   $\beta$  aer mol . Otherwise the authors have to use the rather awkward notation BSR-1 to refer to that ratio and the text may even fall in ambiguities, for example in line 3 of page 8 when they say "Figure 2a shows the simulated aerosol backscatter ratio (i.e. BSR – 1)". But the backscatter ratio is BSR, not BSR-1. Perhaps just /  $\beta$   $\beta$  aer mol would do.

Done. The X-axis of Figure 2a-2b-2c (mode radius) was changed to logarithmic scale, and the information on number concentration ( $N = 10^3 \text{ cm}^{-3}$ ) was added to the caption of Figure 2. The notation 'BSR-1' was replaced with the more compact ' $\beta_{aer}/\beta_{mol}$ ' throughout Section 3.2 and in Figure 2.

**Anonymous Referee #2**

The manuscript shows an intercomparison between a Raman lidar, a commercial ceilometer and an optical in-situ sonde takes as reference. This contribution doesn't represent a substantial contribution to scientific progress and I agree that it is more indicated for Atmospheric Measurement Techniques than ACP.

**Major Issues:**

The FOV correction factor has been computed with a well defined aerosol distribution. How the results change if a more likely bi-modal distribution is used instead? Or changing the distribution width and/or the refractive index? As it is implemented, the correction is depending on a particular type of aerosol.

The single-lognormal size distribution assumed for the calculation of the FOV correction is to be interpreted as an *average* size distribution of boundary layer aerosols, rather than that of a well-defined population. This assumption has the advantage that the correction factors can be described as functions of a single parameter ( $R_m$ ), which can be constrained through the observed AE, as discussed in Section 3.2. Assuming a more complex (e.g. bi-modal) size distribution, as well as relaxing one or more parameters of the size distribution (width, refractive index), would inevitably result in increased number of degrees of freedom of the correction factors, for which insufficient observational constraints are available. Therefore, the correction would be practically not applicable. In particular, for a bi-modal size distribution the correction factors would also depend on the number concentration (N) ratio of the two modes, whereas the correction for a single-lognormal distribution are independent of N.

Since furthermore, previous studies show that a mono-modal distribution represents well the average size distribution of continental aerosols in the Northern mid-latitudes (e.g., Watson-Perris et al., 2019), we believe the assumption of a single lognormal size distribution is justified. This choice is now motivated more carefully in the manuscript (Page 8, Lines 12-15), and a quantitative discussion of the uncertainty introduced by the FOV correction in the statistical comparison is now also provided in Section 4.3 (Page 15, Lines 12-15).

The statistical intercomparison is unclear and counter-intuitive. I would suggest to the authors to use the Pearson Cross Correlation coefficient paired with the Root Mean Square Error, on the whole atmospheric profile and at different altitude ranges, e.g. into the PBL, free troposphere...

Following this comment and one remark of Reviewer #2, the statistical comparison (including mainly Section 4 and Figures 5-10) has been strongly revised and improved. In particular, the following changes were made:

- Low and medium-high aerosol content measurements are separated according to an empirical threshold, and shown by different colors in Figures 5, 7, 8, 10.
- Mean and standard deviations profiles (Figures 5, 8) are calculated from medium-high aerosol content only, and the use of relative differences in the discussion is mainly restricted to medium-high aerosol content measurements.
- The probability density functions (Figures 6, 9) are calculated for medium-high aerosol content and all data separately.
- The Pearson correlation coefficient is now also calculated for medium-high aerosol content data, allowing to quantitatively evaluate the linearity of the correlation between the lidars and COBALD measurements.
- A new Section 3.3 (Page 9 Line 10 Page 10 Line 17) was added to the manuscript, where we introduce the sorting of the data according to aerosol content, and formally define the compared quantities (Equations 3-6).

This approach allows to meaningfully quantify the deviations of medium-high aerosol content measurements at all altitudes, while at the same time not fully neglecting low aerosol content measurement in the free troposphere, and maintaining a clear and systematic structure. The Pearson correlation coefficient, evaluated for medium-high aerosol content data in three altitude intervals (0.8-3 km, 3-6 km, 0.8-6 km), adds a useful further insight to the characterization of the performances of the lidar instruments with respect to COBALD.

Given the similarity between the definitions of RSME and standard deviation (see Equation 5 in the manuscript), and the fact that standard deviation is predominantly used in previous aerosol backscatter intercomparison in the context of EARLINET (e.g., Matthais et al., 2004; Pappalardo et al., 2004), here we decide to keep standard deviation as a measure of variability. This choice also aims to avoid redundancy of information, which might turn out confusing for a reader (note that already three statistical parameters are used in this paper: mean deviation, standard deviation, and Pearson correlation coefficient, each of them quantified for different datasets and different altitude regions).

We believe the revised statistical comparison is strongly improved compared to the previous version of the paper, both in terms of scientific content and clarity, and we thank the reviewer for this comment.

Figure 5 and 8 are unnecessary.

Unfortunately, we are unable to understand properly this comment. Figures 5 and 8 show the vertical profiles of the RALMO – COBALD (Figure 5) and CHM15K – COBALD (Figure 8) difference in aerosol backscatter coefficient, which are a fundamental component of our statistical comparison. As discussed in the previous answer, Figures 5 and 8 were strongly improved in the revised version of the manuscript.

CHM15K inversion should be explained more in detail, as the molecular signal can be very low at 1064nm.

Using a similar ceilometer (CHM15kx by Jenoptik, Germany), Wiegner and Geiss (2012) show that a Klett inversion algorithm can provide accurate aerosol backscatter profiles, despite the low molecular backscatter at infrared wavelengths and the low signal-to-noise ratio in the free troposphere. This reference is now included in the manuscript (Page 4, Lines 11-13).

Optical measurements are strongly affected by water vapor absorption at 940nm. Usually, the ceilometers at this wavelength use a radiative transfer computation to correct the profile. What about COBALD sonde? In the text it is not mentioned.

The effect of water vapor absorption on COBALD BSR at 940 nm is negligible, due to the short optical path length of this instrument ( $\approx$  10 m). COBALD uses two LEDs emitting 250 mW optical power and a detector with FOV of 6°. A good overlap between the emitted light beams and the detector FOV is established at  $\approx$  0.5 m distance from the sonde, and the backscattered signal from a distant layer decreases by the inverse distance squared. This means that, assuming uniform scattering conditions, the signal contribution at 10 m distance from the sonde falls below 0.25 % compared to that in the vicinity of the detector. In this sense, COBALD is considered as an *in-situ* instrument in this comparison.

Therefore, the uncertainty contribution associated with water vapor absorption at 940 nm on COBALD BSR is to be considered as included in the estimate of 5 % accuracy and 1 % precision provided by Vernier et al. (2015), now also given in the manuscript (Page 5, Lines 7-8).

Moreover, some equations are needed to better explain lines 25-28 (Pag. 4).

A new equation was added to define BSR (Equation 1), and the explanation has been broken in multiple sentences for more clarity (Page 5, lines 3-8).

It would be more interesting to intercompare the two instruments vs. COBALD for different meteorological conditions and aerosol loading.

We agree only in part with this comment. The performances of the analyzed instruments, both the lidars and COBALD, are to our best knowledge independent of meteorological conditions, hence we do not expect any systematic bias associated with meteorological parameters (such as temperature or specific humidity). Furthermore, our comparison already avoids all in-cloud and high RH measurements. This is done partly through the selection of the dataset (discussed in Section 2.3), where profiles for which a precise calibration of the lidar signal cannot be achieved are rejected (most frequently due to fog or low clouds), and subsequently through the rejection of all data points with RH > 90% in the statistical comparison (according to the radiosonde measurements) (see Page 7, Lines 1-3). Finally, the limited available dataset (17 soundings for RALMO vs. COBALD, 31 for CHM15K vs. COBALD), together with the irregular periodicity of COBALD soundings (see Table S1 in Supplementary material), and the strong day-to-day variability of boundary layer aerosols, do not allow for any robust investigation of seasonal patterns or other systematic weather-related behaviors.

On the other hand, our comparison does already take different aerosol loadings into account. This is done through the scatter plots of RALMO vs. COBALD  $\beta_{aer}$  (Figure 7) and CHM15K vs. COBALD  $\beta_{aer}$  (Figure 10), where their deviations are displayed and discussed as functions of the absolute COBALD  $\beta_{aer}$  signal, i.e. the aerosol loading (see in particular Page 13 Lines 15-18, Page 14 Lines 26-30). In the revised version of the manuscirpt, this aspect is further emphasized by the distinction between low and medium-high aerosol content measurements, now shown in different colors in Figures 5, 7, 8, 10, and evaluated separately in the statistical comparison (see Sections 3.3, 4.2, 4.3, and replies to comments above).

**Validation of aerosol backscatter profiles from Raman lidar and ceilometer using balloon-borne measurements**

Simone Brunamonti1\*, Giovanni Martucci1, Gonzague Romanens1, Yann Poltera2, Frank G. Wienhold2- Maxime Hervo1, Alexander Haefele1 and Francisco Navas-Guzmán1

1Federal Office of Meteorology and Climatology (MeteoSwiss), Payerne, Switzerland
 2Swiss Federal Institute of Technology (ETH), Zürich, Switzerland
 \*Now at: Swiss Federal Laboratory of Material Sciences and Technology (Empa), Dübendorf, Switzerland

10 Correspondence to: Francisco Navas-Guzmán (francisco.navasguzman@meteoswiss.ch)

5

**Abstract.** Remote sensing measurements by light detection and ranging (lidar) instruments are fundamental for the monitoring of altitude-resolved aerosol optical properties. Here, we validate vertical profiles of aerosol backscatter coefficient ( $\beta_{aer}$ ) measured by two independent lidar systems using co-located balloon-borne measurements performed by Compact Optical Backscatter Aerosol Detector (COBALD) sondes. COBALD provides high-precision in-situ measurements of  $\beta_{aer}$  at two wavelengths

- 15 ter Aerosol Detector (COBALD) sondes. COBALD provides high-precision in-situ measurements of  $\beta_{aer}$  at two wavelengths (455 and 940 nm). The two analyzed lidar systems are the research Raman Lidar for Meteorological Observations (RALMO) and the commercial CHM15K ceilometer (Lufft, Germany). We consider in total 17 RALMO and 31 CHM15K profiles, colocated with simultaneous COBALD soundings performed throughout the years 2014-2019 at the MeteoSwiss observatory of Payerne (Switzerland). The RALMO (355 nm) and CHM15K (1064 nm) measurements are converted to respectively 455 nm
- 20 and 940 nm using the Angstrom exponent profiles retrieved from COBALD data. To account for the different receiver field of view (FOV) angles between the two lidars (0.01-0.02°) and COBALD (6°), we derive a custom-made correction using Mietheory scattering simulations...Our analysis shows that both RALMO and CHM15Klidar instruments achieve a on average a good agreement with COBALD measurements in the boundary layer and free troposphere, up to 6 km altitude, and including fine structures in the aerosol's vertical distribution. For medium-high aerosol content measurements at altitudes below 3 km,
- 25 the mean ± standard deviation difference in βaer calculated from all considered soundings is 2 % ± 37 % (– 0.018 ± 0.237 Mm-1 sr-1 at 455 nm) for RALMO COBALD, and + 5 % ± 43 % (+ 0.009 ± 0.185 Mm-1 sr-1 at 940 mm) for CHM15K COBALD. Above 3 km altitude, absolute deviations generally decrease while relative deviations increase, due to the prevalence of air masses with low aerosol content. Uncertainties related to the FOV correction and spatial and temporal variability effects (associated with the balloon's drift with altitude and different integrations times) contribute to the large standard devi-
- 30 ations observed at low altitudes. The lack of information on the aerosol size distribution and the high atmospheric variability prevent an accurate quantification of these effects. Nevertheless, the excellent agreement observed in individual profiles, including fine and complex structures in the  $\beta_{aer}$  vertical distribution, shows that, that under optimal conditions, the discrepancies between the two instruments with the in-situ measurements are typically are typically smaller than comparable to the the average of our statistical comparison estimated statistical uncertainties of the remote sensing measurements, the excellent agreement

1

observed in individual soundings shows that optimal conditions the discrepancies between the two instruments are typically much smaller than the standard deviations of our statistical comparison. Therefore, we conclude that  $\beta_{aer}$  profiles measured by the RALMO and CHM15K lidar systems are in good agreement with in-situ measurements by COBALD sondes up to 6 km altitude.

- 5 For altitudes below 2 km, the mean  $\pm$  standard deviation difference in  $\beta_{eeer}$  is + 6 %  $\pm$  40 % (+ 0.005  $\pm$  0.319 
[revised manuscript text omitted]

**3.32. Field of View (FOV) correction**

Besides their wavelengths, the COBALD and lidar systems differ in terms of field of view (FOV) of their respective receivers.
RALMO and CHM15K use highly focused laser beams, and consequently have narrow FOVs (200 µrad and 450 µrad, respectively, corresponding to 0.01-0.02°), while COBALD's photodiode detector has a macroscopic FOV of 6° (see Table 1). Considering that the Mie-scattering phase function, i.e. the distribution of scattered light with angle by a spherical particle, has a local maximum in the backward direction (180°), it follows from its wider FOV that COBALD will measure less backscattered radiation (namely, the average intensity between 174°-180°) compared to the lidars (≈ 180°).

To quantify this effect, we performed idealized Mie-theory scattering simulations using the optical model by Luo et al. (2003). We assume a single lognormal size distribution of aerosol particles characterized by mode radius  $R_m$ , number concentration N, fixed width ( $\sigma$ -standard deviation 1.4), and refractive index (1.4). Then, the The BSR of this population is then computed both assuming the phase function value at 180°, corresponding to the lidar observations (FOV  $\approx$  0°), and taking the average of

- 5 the phase function between angles  $174^{\circ}$ - $180^{\circ}$ , corresponding to the COBALD measurements (FOV = 6°). The results are presented in Figure 2. The use of a mono-modal size distribution with fixed width has the advantage that the correction factors can be described as functions of a single parameter ( $R_m$ ), which can be constrained through the observed AE. Furthermore, amono-modal distribution represents well the average size distribution of continental aerosols in the Northern mid-latitudes (e.g., Watson-Perris et al., 2019)
- 10 Figure 2a shows the simulated ratio of aerosol-to-molecular backscatter coefficient,  $\beta_{aer}$  aerosol- $\beta_{mol}$ -backscatter ratio (i.e., BSR = 1: see Equation 1) at 455 nm (blue) and 940 nm (red), as function of  $R_m$  (40 nm-4 µm), calculated assuming FOV  $\approx$ 0° (solid lines) and FOV = 6° (dashed lines), as function of  $R_m$  for the interval 10 nm - 4 µm, and  $N = 10^3$  cm-3. As expected, the simulations show that for all mode radii the COBALD  $\beta_{aer}$  BSR-is lower than the here BSR-measured by the lidar instruments (Figure 2a). Figure 2b shows the lidar-to-COBALD ratio of  $\beta_{aer}$  BSR - 1 (ratio of solid-to-dashed curves in Figure
- 15 2a), i.e. the correction factor required to compensate for this the FOV effect, for 455 and 940 nm as function of  $R_{m^2}$  (note that  $\beta_{aer}$  is proportional to BSR 1, so that this ratio corresponds to the correction factor for  $\beta_{aer}$ ). For the considered size interval of mode radii, the correction factors vary between approximately 1-1.5 and show a non-linear dependency on  $R_m$ , with a local maximum near  $R_m \approx 800$  nm ( $\lambda = 455$  nm) and  $R_m \approx 1.6 \mu$ m ( $\lambda = 940$  nm). This complex optical behavior needs to be corrected. Note that the correction factors in Figure 2b are independent of N, unlike the  $\beta_{ner}/\beta_{mod}$  ratios BSR in Figure 2a.
- To account for the size-dependency in Figure 2b, we use the AE as an indicator of particle size, and develop a parametrization of the correction factors based on the AE measured from COBALD. Figure 2c shows AE between 455-950 nm calculated from the Mie simulations, as function of  $R_m$ . The AE decreases non-monotonically with mode radius and exhibits the characteristic Mie oscillations in the range of approximately  $20 \cdot 40$  nm 1  $\mu$ m (Figure 2c). More in detail, we observe that AE > 1.5 corresponds to small particles ( $R_m < 75$  nm), AE < 0.8 to large particles ( $R_m > 1.16 \mu$ m), while 0.8 < AE < 1.5 corresponds to 75
- 25 nm  $< R_m < 1.16 \mu$ m, but in this intermediate range the change of AE with  $R_m$  is not monotonic (Figure 2c), hence a one-to-one correspondence cannot be established. To simplify this behavior, we choose to-parametrize the correction factors within the three fixed intervals of AE just introduced, and for each interval of AE we take the average correction factor in the corresponding interval of  $R_m$ . Hence, for all measurements with 0.8 < AE < 1.5 we apply the average correction factors between 75 nm 1.16  $\mu$ m (namely, 1.23 at 455 nm, 1.10 at 940 nm) to all measurements with 0.8 < AE < 1.5), for AE < 0.8 the average
- 30 correction factors between 1.16 4  $\mu$ m (1.29 at 455 nm, 1.28 at 940 nm) for AE < 0.8, and no correction for AE > 1.5 (both correction factors  $\approx$  1 for  $R_m$  < 75 nm) we do not apply any correction (both correction factors  $\approx$  1 for  $R_m$  < 75 nm). The resulting FOV correction as function of AE is shown in Figure 2d.

The FOV correction shown in Figure 2d is applied to all COBALD measurements in the statistical comparison. Since, for every AE, the correction factors are larger for 455 nm than for 940 nm (Figure 2d), the FOV correction will affect the RALMO

Formatiert: Englisch (Vereinigtes Königreich) Formatiert: Englisch (Vereinigtes Königreich) Formatiert: Englisch (Vereinigtes Königreich) comparison more than for the CHM15K one. We note that, due to the variability of AE observed in our dataset (see Figure \$2 \$1 in Supplementary material), the middle interval of the correction (0.8 < AE < 1.5) accounts for the large majority of data points in the PBL, AE > 1.5 typically corresponds to free tropospheric background measurements, which are unaffected by the correction, while values of AE < 0.8, corresponding to very large particles, are rarely encountered in our dataset. The effect of

5 the FOV correction on two individual selected profiles and the statistical comparison is will be discussed further in the next sSection 4.1.

**3.3. Compared quantities**

25

After the wavelength conversion and the application of the FOV correction, z

- 10 the difference  $(\frac{\delta z}{\delta t})$  in aerosol backscatter coefficient  $(\Delta \beta_{aer})$  between the lidars *(LID)* and COBALD *(COB)* is calculated for  $\frac{\delta z}{\delta t}$  every each sounding and every each-altitude level-  $(z_i)$  (as in Equation 23):. The mean deviation  $(\delta)$  in given altitude layer and of a given subset of data is calculated according to Equation 4, where  $z_1, \dots, z_N$  represents is the ensemble of all vertical levels in the considered profiles dataset and altitude region. The spread of the individual differences around  $\delta$  is quantified using standard deviation  $(\sigma)$ , defined by Equation 5. Mean and standard deviations are expressed both in absolute backscatter
- 15 coefficient units (Mm-1 sr-1) and in percent units relative to the COBALD signal.

$$\Delta \beta_{aer}(z_i) = \beta_{aer}^{LID}(z_i) - \beta_{aer}^{COB}(z_i)$$
(Equation 3)

ensemble of is calculated according to Equation 3

$$\delta = \frac{\sum_{i=1}^{N} \Delta \beta_{aer}(z_i)}{\Delta N}$$

(Equation 4)

20 whote that the same definition applies for the calculation of the average difference at a given altitude level between all soundings, as well as for the average for a single sounding).

.

σ

54)

The variability of all deviations with respect to  $\Delta$  in the considered ensamble is quantified throught the standard deviation  $(\rho)$ , calculated according to Equation 4:

$$= \sqrt{\sum_{i=1}^{N} (\Delta \beta_{aer}(z_i) \delta_i - \delta \Delta)^2}$$
(Equation

|    | In a typical aAtmospheric backscatter profiles, are typically characterized by a -large gradient in $\beta_{aer}$ between the boundary                                                                                     | Formatiert: Abstand Vor: 0 Pt.                                                         |          |
|----|----------------------------------------------------------------------------------------------------------------------------------------------------------------------------------------------------------------------------|----------------------------------------------------------------------------------------|----------|
|    | layer, with high aerosol content (hence high $\beta_{aer}$ ), and the free troposphere, with low aerosol content (low $\beta_{aer}$ ). This gradient                                                                       |                                                                                        |          |
|    | is such that the same absolute $\Delta\beta_{aer}$ may correspond to either a small or large relative $\Delta\beta_{aer}$ , depending on altitude. In particular,                                                          | Formatiert: Schriftart: Kursiv                                                         |          |
|    | free tropospheric measurements, where statistical fluctuations often dominate over the atmospheric signal, typically yield large                                                                                           |                                                                                        |          |
| 5  | relative deviations in spite of small absolute differences. While the boundary layer is the main region of the interest of this                                                                                            |                                                                                        |          |
|    | study, as it contains most of the aerosol loading in the column, the free troposphere (including low aerosol content measure-                                                                                              |                                                                                        |          |
|    | ment) cannot be completely neglected, since a good agreement at high altitudes ensures that all profiles are well calibrated (see                                                                                          |                                                                                        |          |
|    | Section 2.3). Therefore, here we focus our analysis on medium-high aerosol content data (defined as explained below), yet for                                                                                              |                                                                                        |          |
|    | completeness we also display low aerosol content measurements in the statistical comparison. all                                                                                                                           |                                                                                        |          |
| 10 | The aerosol content is evaluated according to the average COBALD $\beta_{aer}$ in each profile and 300 m altitude interval (i.e., mean                                                                                     |                                                                                        |          |
|    | of 10 vertical levels). Based on the observed range of variability of $\beta_{aer}$ in our dataset (see Figures S1 supplementary material),                                                                                |                                                                                        |          |
|    | we define 'low aerosol content' all layers with average COBALD $\beta_{aer} < 0.1 \text{ Mm}^{-1} \text{ sr}^{-1}$ at 455 nm (RALMO comparison), and                                                                       |                                                                                        |          |
|    | average COBALD $\beta_{aer} < 0.05 \text{ Mm}^{-1} \text{ sr}^{-1}$ at 940 nm (CHM15K comparison). The averaging in 300 m layers ensures that actual                                                                       |                                                                                        |          |
|    | air masses with low aerosol content are identified, rather than individual data points exceeding the threshold due to statistical                                                                                          |                                                                                        |          |
| 15 | variability. When the above conditions are met, all data points in the considered layer are classified as 'low aerosol content'.                                                                                           |                                                                                        |          |
|    | All other data points are referred to as 'medium-high aerosol content'. Note that this definition allows individual data points                                                                                            | Formatiert                                                                             |          |
|    | to exceed the threshold, as long as the average criteria in the layer are not exceeded.                                                                                                                                    |                                                                                        |   |
|    | For medium-high aerosol content data,                                                                                                                                                                                      | Formatiert: Block, Einzug: Links: 0 cm, Erste Zeile: 0                                 |          |
|    | Finally in addition to $\delta$ and $\sigma_{a}$ we also evaluate ,-the correlation between the lidars the lidars and and COBALD is evaluated                                                                              | cm, Abstand Vor: 0 Pt., Nach: 0 Pt.                                                    |          |
| 20 | throughusing the Pearson correlation coefficient $(\rho)_{re}$ This is defined defined according to Equation 56, where $B_{LID}$ and $B_{COB}$                                                                             | Formatiert                                                                             |          |
|    | are respectively the average lidar $\underline{\beta_{aer}}$ and COBALD $\underline{\beta_{aer}}$ , calculated in 300 m layers. The Pearson correlation coefficient rep-                                                   |                                                                                        |          |
|    | resents the degree of linearity of the correlation between $\beta_{aer}^{LID}$ and $\beta_{aer}^{COB}$ , ranging between values of -1 (total negative linear                                                               | Formatiert                                                                             |          |
|    | correlation) and +1 (total positive linear correlation). In the statistical comparison, $\beta$ , $\rho$ , and $\rho$ are quantified for both RALMO                                                                        |                                                                                        |          |
|    | and CHM15K in three altitude intervals of 0.8-3 km asl, 3-6 km asl, and 0.8-6 km asl (i.e., all altitudes). +                                                                                                              |                                                                                        |          |
Abstand Vor: 0 Pt., Nach: 0 Pt. |          |
|    | $\rho = \frac{\sum_{i=1}^{N} \left( \beta_{aer}^{LID} (\lambda_{z_i}) - B_{LID} \right) \cdot \left( \beta_{aer}^{COB} (\lambda_{z_i}) - B_{COB} \right)}{\left( \beta_{aer}^{COB} (\lambda_{z_i}) - \beta_{COB} \right)}$ | Formatiert: Einzug: Links: 2.5 cm, Erste Zeile: 1.25 cr                                | n,       |
|    | $\sqrt{\sum_{i=1}^{N} \left(\beta_{aer}^{LID}(\lambda_{zi}) - B_{LID}\right)^2 \cdot \sum_{k=1}^{N} \left(\beta_{aer}^{COB}(\lambda_{zi}) - B_{COB}\right)^2}$                                                             | Abstand Vor: 12 Pt.                                                                    | $ \_ $   |
|    | (Equation 56)                                                                                                                                                                                                              | Formatiert: Schriftart: 11 Pt.                                                         |          |
|    | Where $B_{\mu\nu}$ and $B_{cons}$ are the average lidar and COBALD $\beta_{aer}$ in the considered                                                                                                                         | Formatiert                                                                             |          |
|    | ensemble of data: $B_{LHP} = \frac{\sum_{k=1}^{N} \beta_{kHP}^{t+IP}(\lambda, z_k)}{N}$ (Equation                                                                                                                          | Formatiert: Einzug: Links: 4.99 cm, Erste Zeile: 1.25 cm, Abstand Nach: 6 Pt.          |          |

**(Equation 7)**

The Pearson correlation coefficient represents the degree of linear correlation of  $\beta_{aeer}$  between the lidars and COBALD, ranging between values of -1 (total negative linear correlation) to +1 (total positive linear correlation). Mean and standard deviations are expressed in absolute backscatter coefficient values (units of Mm+-sr+) as well as in percent units relative to the COBALD signal (Equations 8, 9):

 $\Delta_{pel} = \frac{1}{N} \sum_{i=1}^{N} \frac{\delta_i}{\rho_{core}^{COB}(\rho_{el})} - (Equation 8)$   $\sigma_{ret} = \sqrt{\frac{1}{N-1} \sum_{i=1}^{N} \frac{(\delta_t - \Delta)^2}{\rho_{core}^{COB}(\rho_t)}} - (Equation 9)$

**A common issue when comparing aerosol backscatter coefficients**

[revised manuscript text omitted]

**4.2.1. RALMO vs. COBALD**

Figure 5 shows all data points of the RALMO – COBALD difference ( $\Delta \beta_{aer}$  at 455 nm) as function of altitude, both expressed in absolute backscatter coefficient units (Panel a) and in percent units relative to the COBALD signal (Panel b), after the FOV correction was applied to all COBALD measurements. Medium-hHigh and Low-low aerosol content measurements, classified as in Section 3.3, are shown by dark blue and light blue circles, respectively. The mean deviation ( $\hat{\rho}$ ) and mean ± standard deviation ( $\hat{\delta} \pm \sigma$ ) deviation profiles of Mediummedium-High-high aerosol content data data of  $\Delta \beta_{wer}$ -are shown in both panels

as shown by thick solid and thin solid dashed-black lines, respectively, respectively. As discussed in Section 3.1, to avoid incloud measurements, we only consider data points with RH < 90% (according to the radiosonde measurements). Medium-high aerosol content measurements of RALMO and COBALD  $\beta_{aer}$  measurements a are on average in good agreement over the entire altitude range (0.8–0.8–6 km asl), yet significant discrepancies can occur in single individual profiles, and the

- 5 standard deviation is not constant with altitude. As expected, the largest absolute differences are observed For ε > 2.5 km, typically corresponding to the free troposphere (i.e. above the PBL), the absolute differences between RALMO and COBALD are small (Figure 5a), while their relative differences are large (Figure 5b). This is mostly due to the low acrosol content, hence the low absolute backscattered signal, in 'clean' free-tropospheric air masses. The absolute *Aβouv*-differences often exceed ± 100% of the signal, and the mean *Aβouv* profile varies between 0-30 % (Figure 5b).
- Fat low altitudes ( $z \le -3z \le 2.5$  km), which approximately corresponds to the average top of including most of the PBL (hence PBL altitude medium-high aerosol content) measurements in our dataset (Figure 5a). Converselyin our dataset, smaller absolute discrepancies, yet large relative differences (Figure 5b), are found in the free troposphere (z > 3 km), where low aerosol content measurements prevail.
- 15 For z < 2.53 km, the discrepancies between RALMO and COBALD are larger in absolute terms (Figure 5a), but smaller in relative terms (Figure 5b) compared to the free troposphere, the mean  $\Delta\beta_{mer}$  deviation profile ( $\partial$ ) of Medium Highmedium-high aerosol content data stays within  $\pm$  0.1 Mm-1 sr-1, with while standard deviation ( $\underline{a}$ ) ~ ranges between 0.1-0.425 Mm-1 sr-1, while and individual data points rarely exceed  $\pm$  0.5 Mm-1 sr-1 (Figure 5a). In relative terms,  $\underline{b}$  shows an average slight overestimation of 5-10-10 % below 2 km (with  $\underline{a} \approx 4040$  %), and an underestimation of 10-25 % between 2-3 km ( $\underline{a} \approx 4030$  %)
- 20 (Figure 5b). A large fraction of this high variability can be attributed to the atmospheric variability effects discussed in Section 3.1Such aSuch relatively large relative standard deviations can be at least partly attributed to, discussed in see Figures 3 4, S3-S42 the uncertainties associated with the wavelength conversion and FOV correction of the data (Sections 3.1-3.2) and spatial and temporal variability effects (Section 2.4). Theise issue will be discussed more in more detail in Section 54.3. For z > 2.53 km+, typically corresponding to the free troposphere (i.e. above the PBL), the absolute differences between
- 25 RALMO and COBALD are small (Figure 5a), while their relative differences are large (Figure 5b). This is mostly due to the low aerosol content, hence the low absolute backscattered signal, in 'clean' free tropospheric air masses. Thenearly all the observedabsolute absolute Δβaer differences for z > 2.5 km are smaller than ± 0.1 Mm-1 sr-1 for the majority of data points (Figure 5a), yet their relative discrepancies often exceed ± 100% of the signal, and the mean Δβaer profile varies between 0.30 % (Figure 5b). Medium High Medium-high aerosol content data points above 3 km altitude are generally mostly foundstay.
- 30 within-a deviations of  $\pm$  50 % -discrepancy, whereas low aerosol content ones often exceed  $\pm$  100 % -(Figure 5b).

To quantify the spread of  $\Delta \beta_{aec}$ , Figure 6 shows the frequency of occurrence distribution of the RALMO – COBALD  $\Delta \beta_{aec}$ , difference, calculated for the altitude intervals of 0.8-30–2 km (Panels e-fa-b),  $\frac{2-43-6}{2}$  - km (Panels c-d) and  $\frac{800 \text{ m} - 60.8}{2}$ .

[revised manuscript text omitted]

Table 2. Overview of the sStatistical comparison of of RALMO vsvs.2 COBALD: results for medium-high aerosol content (17 profiles, 2016-2019). For each altitude intervalFor each data interval, we show mean and standard deviationdeviation ( $\phi$ , both in absolute units and percent units relative to COBALD) and standard deviation ( $\sigma$ , both in absolute units and percent units relative to COBALD) and standard deviation ( $\sigma$ , both in absolute units and percent units relative to COBALD) at 455 nm, and of the RALMO - COBALD difference in aerosol backscatter coefficient ( $A\beta_{uer}$ ) at 455 nm, calculated both before (left) and after (right) the FOV correction was applied to the COBALD dataPearson correlation coefficient ( $\phi$ ). Note that the data intervals are the same as those selected for the frequency of occurrence distributions shown in Figure 5 (2-km altitude bins between 0-6 km) and Figure 6 (all data points with COBALD  $\beta_{uer} > 0.2 \text{ Mm}^{-1} \text{ sr}^{-1}$ ,

| CHM15K  | CORALD | (21) | nrofilog | 2014 | 2010  |
|---------|--------|------|----------|------|-------|
| CHINISK | CODALD | (JI  | promes   | TOLL | 20177 |

|                                                         | Before FOV correction                      |                                            | After FOV correction                       |                                             | Formatiert: Englisch (Vereinigte Staaten) |
|---------------------------------------------------------|--------------------------------------------|--------------------------------------------|--------------------------------------------|---------------------------------------------|-------------------------------------------|
| Interval                                         | <del>Mean Δβeer(940 nm)</del>   | Standard deviation                         | <del>Mean Δβrer(940 nm)</del>   | Standard deviation                          | Formatiert: Englisch (Vereinigte Staaten) |
| $0.9 < \pi < 2 \text{ km} = 1$                          | + 0.089 Mm -+ sr -+  | ± 0.211 Mm -+ sr -+  | + 0.038 Mm -+ -sr -+ | ±0.207 Mm -+ -sr -+   | Example of Englisch (Vereinigte Staten)   |
| 0.0 < z < 2  km a.s.i.                                  | <del>(+ 25.2 %)</del>                      | <del>(± 53.6 %)</del>                      | <del>(+ 13.1 %)</del>                      | <del>(± 51.0 %)</del>                       |                                           |
| 0                                                       | + 0.007 Mm -1 -sr -1 | ± 0.150 Mm -1 -sr -1 | = 0.013 Mm -1 -sr -1 | $\pm 0.162 \text{ Mm}^{-1} \text{ sr}^{-1}$ |                                           |
| z < z < 4 km a.s.1.                                     | <del>(-3.71 %)</del>                       | <del>(± 322 %)</del>                       | <del>(- 8.44 %)</del>                      | <del>(± 320 %)</del>                        |                                           |
|                                                         | -0.015 Mm -1 sr -1   | ± 0.119 Mm + sr +    | $= 0.020 \text{ Mm}^{-1} \text{ sr}^{-1}$  | ±0.139 Mm -1 sr -1    |                                           |
| 4 < z < 6  km a.s.l.                                    | <del>(-21.7 %)</del>                       | <del>(± 644 %)</del>                       | <del>(- 23.5 %)</del>                      | <del>(± 641 %)</del>                        |                                           |
| 0 0 0 0 0 1 1                                           | + 0.083 Mm -+ -sr -+ | ± 0.275 Mm -+ -sr -+ | $= 0.001 \text{ Mm}^{-1} \text{ sr}^{-1}$  | ±0.281 Mm -+ -sr -+   |                                           |
| $\beta_{\text{rer}} > 0.1 \text{ Mm}^{+} \text{sr}^{+}$ | <del>(+ 16.2 %)</del>                      | <del>(± 43.2 %)</del>                      | <del>(+ 1.93 %)</del>                      | <del>(± 42.5 %)</del>                       | Formatiert: Englisch (Vereinigte Staaten) |

**CHM15K - COBALD (31 profiles, 2014-2019, Medium-high aerosol content)**

| Altitude interval | Mean deviation (δ)                 | Standard deviation (σ)           | Correlation coefficient (p) |
|--------------------------|-------------------------------------------|-----------------------------------------|------------------------------------|
| 0.8 2 loss and           | +0.009 Mm -1 sr -1  | 0.185 Mm -1 sr -1 | . 0.72                             |
| 0.8 - 3 km asi           | (+ 5.2 %)                          | (43.0 %)                         | +0.72                              |
| 2 have a characteristic  | - 0.081 Mm -1 sr -1 | 0.219 Mm -1 sr -1 | . 0.24                             |
| 3 Km – 6 Km asi   | (-43.3 %)                          | (71.9 %)                         | + 0.24                             |
| 0.8- 6 km asl     | - 0.043 Mm -1 sr -1 | 0.205 Mm -1 sr -1 | .0.62                              |
| (i.e., all altitudes)    | (- 22.6 %)                         | (59.6 %)                         | +0.02                              |

Table 3. Statistical comparison of CHM15K vs. COBALD: results for medium-high aerosol content. For each altitude interval, we show mean deviation ( $\delta$ , both in absolute units and percent units relative to COBALD) and standard deviation ( $\sigma$ , both in absolute units and percent units relative to COBALD) at 940 nm, and Pearson correlation coefficient ( $\rho$ ).

Table 3. Overview of the statistical comparison of CHM15K vs. COBALD (31 profiles, 2014-2019). For each data interval, we show mean and standard deviation (both in absolute units and percent units relative to COBALD) of the CHM15K - COBALD difference in aerosol backscatter coefficient ( $\Delta f_{sec}$ ) at 940 nm, calculated both before (left) and after (right) the FOV correction was applied to the COBALD data. Note that the data intervals are the same as those selected for the frequency of occurrence distributions shown in Figure 8 (2 km altitude bins between 0-6 km) and Figure 9 (all data points with COBALD  $\beta_{sec}$  > 0.1 Mm-t-sr+), **Formatiert:** Schriftart: 11 Pt., Englisch (Vereinigte Staaten)

---

## Referee Report (RR1)

The authors have responded adequately to my remarks in the previous review, but to my main remark No. 4 in what respects the statistical uncertainty of RALMO measurements. In the revised manuscript the authors state (page 4, lines 2-4 of the revised manuscript): "The mean statistical uncertainties associated with the retrieval of  $\beta_{aer}$  at 355 nm from Raman inversion techniques are typically estimated as 15 % in the PBL (Pappalardo et al., 2004)". However, the reference (Pappalardo et al., 2004) supposed to sustain this uncertainty value does not deal with instruments, but with the performance of algorithms faced to synthetic lidar data mimicking an instrument output. The quoted 15% figure refers to the typical statistical error yielded by the different algorithms when dealing with simulated raw signals coming from a typical atmospheric profile and with a given amount of noise yielding a mean signal-to-noise ratio of ~ 70 in the PBL. The mean statistical uncertainty cannot be based on this reference and the sentence must be removed of modified. I suspect, based on RALMO characteristics, that the typical uncertainty for the retrieval of  $\beta_{aer}$  in the PBL from its nighttime data will be lower than 15%.

Other minor issues are:

1. The authors use throughout the paper the same symbol,  $\Delta\beta_{aer}$ , to denote both the absolute difference and the relative difference between the aerosol backscatter coefficient retrieved from a lidar measurement and from COBALD. But in Eq. (3)  $\Delta\beta_{aer}$  is defined unambiguously as absolute error. I suggest that for relative error another symbol is used. I'm sorry I didn't notice this in my previous review.

2. On page 14, lines 17-18 of the revised manuscript, the authors say, referring to the larger spread of relative  $\beta_{aer}$  differences above 3 km between CHM15K and COBALD compared to the relative differences between CHM15K and RALMO: "This again denotes the lower signal-to-noise ratio of CHM15K with respect to RALMO at high altitudes". But couldn't it be due also to the smaller values of  $\beta_{aer}^{COB}$  in the denominator when computing the relative error?

---

## Author Response (AR2)

**Manuscript acp-2020-294 – Author's replies to reviewers**

*We thank gratefully the editor and two anonymous referees for careful reading and comments. Below are the referee's comments in **black**, and replies from the authors in **blue**. Please note that page and line numbers given below refer to the revised version of the manuscript without tracked-changes.*

**Anonymous Referee #1**

The authors have responded adequately to my remarks in the previous review, but to my main remark No. 4 in what respects the statistical uncertainty of RALMO measurements. In the revised manuscript the authors state (page 4, lines 2-4 of the revised manuscript): "The mean statistical uncertainties associated with the retrieval of βaer at 355 nm from Raman inversion techniques are typically estimated as 15 % in the PBL (Pappalardo et al., 2004)". However, the reference (Pappalardo et al., 2004) supposed to sustain this uncertainty value does not deal with instruments, but with the performance of algorithms faced to synthetic lidar data mimicking an instrument output. The quoted 15% figure refers to the typical statistical error yielded by the different algorithms when dealing with simulated raw signals coming from a typical atmospheric profile and with a given amount of noise yielding a mean signal-to-noise ratio of ~ 70 in the PBL. The mean statistical uncertainty cannot be based on this reference and the sentence must be removed of modified. I suspect, based on RALMO characteristics, that the typical uncertainty for the retrieval of βaer in the PBL from its nighttime data will be lower than 15%.

The reviewer is right that the 15% uncertainty estimated by Pappalardo et al. (2004) only accounts for errors related to the data processing algorithms, and does not represent the mean statistical uncertainty of the instruments. Therefore, as suggested, we removed this statement from the manuscript.

Other minor issues are:

1. The authors use throughout the paper the same symbol, Δβaer, to denote both the absolute difference and the relative difference between the aerosol backscatter coefficient retrieved from a lidar measurement and from COBALD. But in Eq. (3) Δβaer is defined unambiguously as absolute error. I suggest that for relative error another symbol is used. I'm sorry I didn't notice this in my previous review.

Done: a new symbol ($\Delta\beta_{aer}^{rel}$) was introduced to denote the relative differences in aerosol backscatter coefficient. For consistency, new symbols were also introduced to denote the relative mean deviation ($\delta_{rel}$) and relative standard deviation ($\sigma_{rel}$) of $\Delta\beta_{aer}$. These new symbols are now used consistently throughout the manuscript, figures and tables.

2. On page 14, lines 17-18 of the revised manuscript, the authors say, referring to the larger spread of relative $\beta_{aer}$ differences above 3 km between CHM15K and COBALD compared to the relative differences between CHM15K and RALMO: "This again denotes the lower signal-to-noise ratio of CHM15K with respect to RALMO at high altitudes". But couldn't it be due also to the smaller values of $\beta_{aer}^{COB}$ in the denominator when computing the relative error?

It is true that, for a given difference $\Delta\beta_{aer}$, the lower absolute $\beta_{aer}$ signal at 940 nm compared to 455 nm leads to larger relative differences ($\Delta\beta_{aer}^{rel}$) for CHM15K – COBALD than for RALMO – COBALD. The statement has been modified in order to include this consideration: "This is due to the low signal-to-noise ratio of CHM15K at high altitudes, together with the lower absolute $\beta_{aer}$ signal at 940 compared to 455 nm" (Page 14, Lines 17-19).

**Anonymous Referee #2**

I am happy that the authors addressed all the previously raised issues and now the manuscript is ready for publication.

Some typos in the manuscript should corrected.
Some images are not in high-res, with very small fonts hard to read.

Done: a careful grammar check of the entire manuscript was performed, and the quality of the figures was improved. In particular, a larger font size is now used in Figures 2, 3, 4, 7, 10 to improve readability, and the aspect ratio of Figures 5, 7, 8, 10 was optimized in order to enhance the resolution in pdf format. We will make sure during proofreading that all figures are reproduced in high resolution in the final layout of the paper.

I would add in the introduction some references about similar campaigns between ceilometers and lidars, e.g:

Tsaknakis, G., et al. "Inter-comparison of lidar and ceilometer retrievals for aerosol and planetary boundary layer profiling over Athens, Greece." Atmospheric Measurement Techniques 4.6 (2011): 1261-1273.

Madonna, F., et al. "Intercomparison of aerosol measurements performed with multi-wavelength Raman lidars, automatic lidars and ceilometers in the framework of INTERACT-II campaign." Atmospheric Measurement Techniques 11.4 (2018).

Done: references to Tsaknakis et al. (2011) and Madonna et al. (2018) were added to the introduction (Page 2, Line 22) and conclusions (Page 17, Lines 13-14). We thank the reviewer for this comment.